# Mitigating Conversational Inertia in Multi-Turn Agents

**Yang Wan**[1]  **Zheng Cao**[1]  **Zhenhao Zhang**[2]  **Zhengwen Zeng**[3]  **Shuheng Shen**[3]  **Changhua Meng**[3]  **Linchao Zhu**[1]

## Abstract

Large language models excel as few-shot learners when provided with appropriate demonstrations, yet this strength becomes problematic in multi-turn agent scenarios, where LLMs erroneously mimic their own previous responses as few-shot examples. Through attention analysis, we identify **conversational inertia**, a phenomenon where models exhibit strong diagonal attention to previous responses, which is associated with imitation bias that constrains exploration. This reveals a tension when transforming few-shot LLMs into agents: longer context enriches environmental feedback for exploitation, yet also amplifies conversational inertia that undermines exploration. Our key insight is that for identical states, actions generated with longer contexts exhibit stronger inertia than those with shorter contexts, enabling construction of preference pairs without environment rewards. Based on this, we propose Context Preference Learning to calibrate model preferences to favor low-inertia responses over high-inertia ones. We further provide context management strategies at inference time to balance exploration and exploitation. Experimental results across eight agentic environments and one deep research scenario validate that our framework reduces conversational inertia and achieves performance improvements.

## 1. Introduction

The emergence of Large Language Models (LLMs) has revolutionized autonomous task completion, with multi-turn dialogue agents becoming increasingly prevalent following the introduction of paradigms like ReAct (Yao et al.,

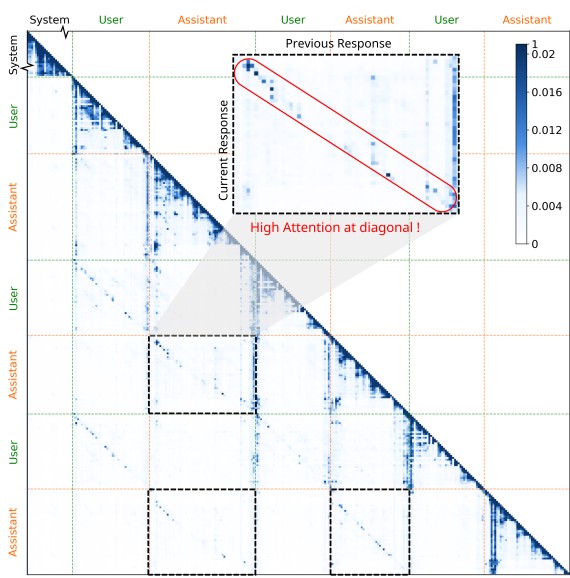

*Figure 1.* Attention visualization in maze environment showing conversational inertia. Despite generating diverse responses, the model exhibits strong diagonal attention to previous assistant outputs, revealing token-to-token correspondence associated with conversational inertia. For other environments, see Appendix R.

2023b). These agents interact with environments through iterative observation-action cycles, accumulating context over extended episodes to solve complex tasks such as web navigation (Yao et al., 2022), embodied AI (Shridhar et al., 2021), and so on.

However, a critical challenge emerges as conversation rounds increase: agent performance degrades significantly even when context remains well within the model's capacity. In the agent domain, more dialogue turns do not necessarily lead to better performance (Hsieh et al., 2024a; Liu et al., 2024; Hong et al., 2025).

To understand this phenomenon, we conduct a detailed analysis of attention patterns in multi-turn agent interactions. Our investigation reveals an issue: *conversational inertia*. Through attention matrix visualization in Figure 1, we discover that models exhibit strong diagonal correspondence when attending to previous assistant responses. Specifically, the $i$-th token in the current response disproportionately

[1]College of Computer Science and Technology, Zhejiang University, Hangzhou, China [2]University of Rochester, Rochester, NY, USA [3]Ant Group, Hangzhou, China. Correspondence to: Linchao Zhu <linchao@zju.edu.cn>.

*Proceedings of the 43rd International Conference on Machine Learning*, Seoul, South Korea. PMLR 306, 2026. Copyright 2026 by the author(s).

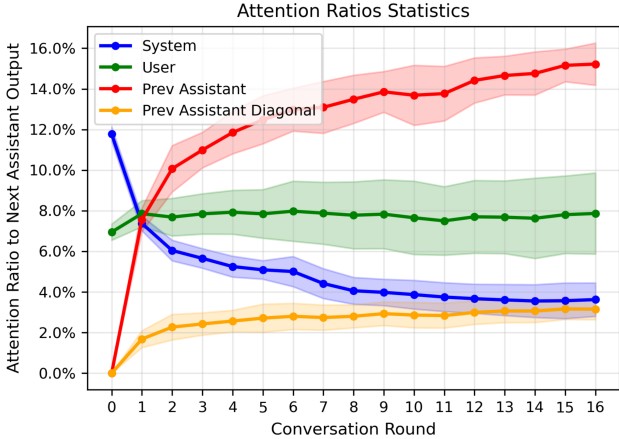

*Figure 2.* Quantitative analysis of each part of attention as context increases, revealing monotonic growth in attention to previous assistant responses while attention allocation to user inputs remains stable. Detailed analysis is presented in Section 3.4.

attends to tokens around the $i$-th position in previous assistant outputs, which is consistent with an imitation bias that accumulates errors and constrains exploration.

This attention pattern has profound implications for agent imitation behavior: Models tend to replicate previous response patterns rather than adapting to new environmental feedback, which is associated with suboptimal decision-making (Laban et al., 2025). Furthermore, as context length increases, attention to system prompts becomes reduced, causing models to rely more heavily on few-shot learning from interaction history rather than following task-specific instructions. Our empirical analysis confirms that this phenomenon represents imitation of homogeneous information rather than simple length-related degradation. As evidence, attention to previous assistant responses grows monotonically with context length, while attention to user inputs remains relatively stable, as shown in Figure 2.

We address this challenge based on a key observation: for identical states, actions generated with shorter contexts exhibit weaker inertia, enabling construction of preference pairs without environment rewards. We propose Context Preference Learning, which calibrates model preferences to favor low-inertia responses by fine-tuning only 0.4% of parameters using long-short context preference pairs. This preference learning enables models to resist inertial degradation, requiring no ground truth environment rewards or expert demonstrations. At inference time, we provide context management strategies that periodically clear interaction history to balance exploration and exploitation. Such periodic context clearing creates breaks at conversation turn boundaries, preventing error propagation by eliminating accumulated inertial patterns.

Extensive evaluation across eight diverse environments and

one deep research scenario validates our approach effectively mitigates conversational inertia. Context Preference Learning reduces diagonal inertia on Qwen3-8B by 11% and achieves approximately 4% performance improvement across all context management approaches. Clip context achieves 4% success rate improvement across all tested models, compared with Window context method. We empirically found reduced inertia from clipping context plays a major role in summarization methods' performance gains, rather than the generated summarization.

The key contributions of this work are: **(1)** identification and analysis of conversational inertia as a contributor to multi-turn agent performance degradation, **(2)** a Context Preference Learning that reduces inertia by calibrating model preferences without requiring environment supervision, and **(3)** a clip context management that achieves better exploration-exploitation balance.

## 2. Method

To mitigate conversational inertia, we propose Context Preference Learning, which calibrates model preferences to favor low-inertia responses using long-short context preference pairs (Section 2.1), and context management strategies that control inertia strength during inference (Section 2.2).

### 2.1. Context Preference Learning

We propose Context Preference Learning (CPL) that exploits the differential inertia between short and long contexts. Our key insight is that for identical states, actions generated with longer input contexts exhibit stronger conversational inertia compared to those with shorter contexts. The long versus window comparison in Table 1 further validates this observation. This enables us to construct **strong-weak inertia preference pairs** without requiring environment-provided reward signals or expert demonstrations.

As illustrated in the left panel of Figure 3, for each state we generate two actions:

- $a^{\text{long}}$ using the full context history (longer context)
- $a^{\text{short}}$ using only recent context (shorter context)

Then, we execute the long-context generated action $a^{\text{long}}$ in the environment to obtain the next observation, which is then used to synchronously update both the long and short context trajectories. This design ensures that the dataset's input context represents lower-quality trajectories, while the preferred chosen actions correspond to higher-quality decisions, encouraging the model to break out of loops in lower-quality input contexts and generate better actions.

The preference dataset uses the shorter context $\mathcal{C}_t^{\text{short}}$ as training input for both chosen and rejected actions. This design

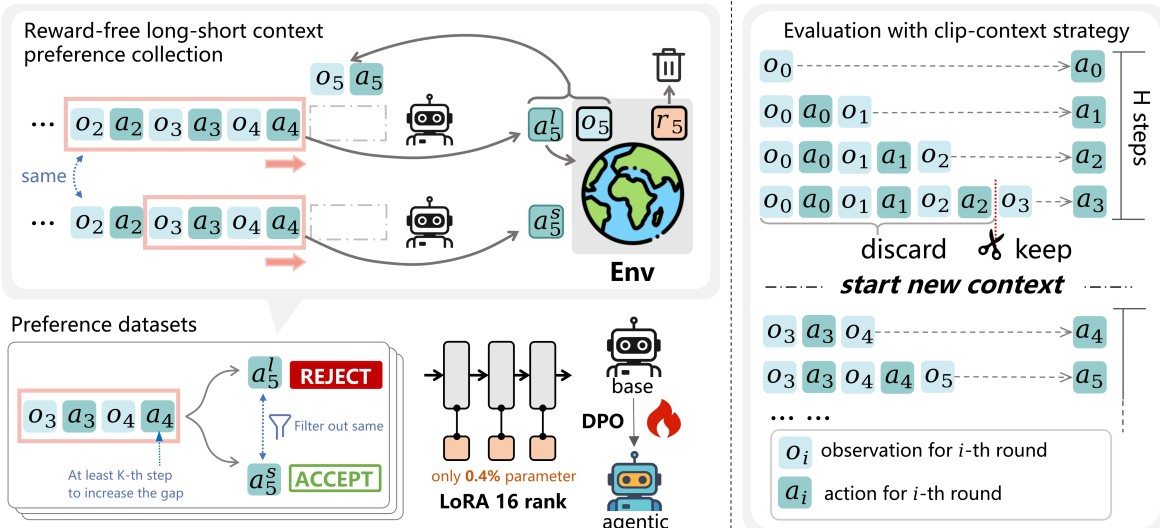

*Figure 3.* **Left**: Context Preference Learning with preference data collection and DPO (Rafailov et al., 2023) training. Our method does not need environment rewards ($r_5$) to generate preference pairs. **Right**: Clip context applied during inference context management.

teaches the model to prefer actions generated from short context $a_t^{\text{short}}$ over those from long context $a_t^{\text{long}}$, which may contain more inertia from accumulated history. By exclusively using short context as input, we prevent the model from learning to rely on the full trajectory history, instead encouraging it to make decisions based on limited but relevant context. To enhance data quality, we implement two filtering strategies: (1) minimum context margin requiring at least $K$ steps before sampling to ensure sufficient context disparity, and (2) action diversity filtering excluding pairs where $a_t^{\text{long}} = a_t^{\text{short}}$ to focus on cases where context length meaningfully impacts decision-making. This preference dataset enables robust preference learning to mitigate inertia even with limited context and training resources.

We train the model using the standard DPO loss:

$$\mathcal{L}_{\text{DPO}} = -\mathbb{E}\left[\log\sigma\left(\beta\log\frac{\pi_\theta(a_t^{\text{short}}|\mathcal{C}_t^{\text{short}})}{\pi_{\text{ref}}(a_t^{\text{short}}|\mathcal{C}_t^{\text{short}})}\right.\right.$$
$$\left.\left. -\beta\log\frac{\pi_\theta(a_t^{\text{long}}|\mathcal{C}_t^{\text{short}})}{\pi_{\text{ref}}(a_t^{\text{long}}|\mathcal{C}_t^{\text{short}})}\right)\right] \quad (1)$$

where $\mathcal{C}_t^{\text{short}}$ is consistently used as input context for both preference options. We employ parameter-efficient LoRA (Hu et al., 2022) fine-tuning for optimization. Detailed training configuration and hyperparameters are provided in Appendix D.1, with LoRA hyperparameter ablations in Appendix D.2

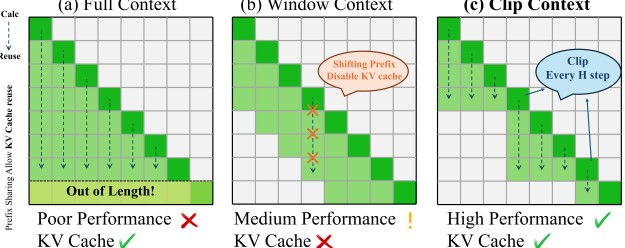

*Figure 4.* Three context control methods and their impact on attention scope. **(a)** Long Context retains complete context but suffers from conversational inertia and theoretically has context capacity limit problems. **(b)** Window Context maintains recent context but cannot leverage KV cache due to shifting boundaries. **(c)** Clip Context (our method) periodically clears context while enabling KV cache optimization.

## 2.2. Inference Context Management

To address conversational inertia and enable agents to break out of error loops, motivated by the finding in Figure 2 that diagonal attention is reduced under shorter history, we introduce Clip Context as a context management solution. We use a unified perspective that encompasses existing context management approaches to compare different context management methods in the multi-turn domain, as illustrated in Figure 4.

Let $\mathcal{C}_t = \{r_1, r_2, \ldots, r_t\}$ denote the conversation context in turn $t$, where each round $r_i$ consists of user input $u_i$ and assistant response $a_i$. For round-wise attention control, we define the attention mask $M_t \in \{0,1\}^{|\mathcal{C}_t| \times |\mathcal{C}_t|}$ that determines which historical rounds are accessible during the generation of current response. We formulate three primary context control strategies.

*Table 1.* Comparison of CPL finetuning and context management methods on eight agent tasks. For fair comparison, we tune optimal hyperparameters for each context management strategy. Improvement in red is relative to Base + Window.

| Model | Method | | Score | | | | | | | | |
|-------|--------|--------|-----|-----|-----|-----|-----|-----|------|-----|---------|
| | **Variant** | **Context Manage** | **MZ** | **ALF** | **WS** | **TC** | **FL** | **HM** | **2048** | **RH** | **Average** |
| Qwen3-8B | Base | Window | 74.0 | 68.2 | 40.0 | 71.3 | **67.5** | 85.3 | 66.9 | 45.8 | 64.9 |
| | | Long | 39.0 | 61.0 | 23.3 | 67.5 | 63.5 | 75.1 | 65.6 | 40.3 | 54.4 |
| | | Summarization | 78.0 | **71.0** | 46.5 | 80.5 | 64.6 | **88.9** | 70.6 | 50.4 | 68.8 |
| | | Clip | **83.0** | 67.7 | 44.4 | 82.3 | 67.5 | 85.1 | 70.9 | 50.2 | **68.9** |
| | CPL | Window | 78.0 | 67.5 | 44.9 | 74.5 | 68.2 | 89.5 | 72.4 | 44.3 | 67.4 |
| | | Long | 44.0 | 59.5 | 40.5 | 65.5 | **70.0** | 77.7 | 70.2 | 44.8 | 59.0 |
| | | Summarization | 87.5 | 71.5 | 52.2 | 75.2 | 69.4 | **89.9** | 71.6 | 49.1 | 70.8 |
| | | Clip | **91.5** | 70.3 | **54.9** | 83.0 | 68.4 | 87.5 | **73.0** | 51.1 | **72.5** +7.6 |
| Llama3.1-8B | Base | Window | 79.7 | 21.5 | **45.4** | 68.2 | 58.6 | 87.0 | **70.1** | 38.0 | 58.6 |
| | | Long | 48.0 | 20.3 | 29.1 | 49.1 | 54.0 | 79.9 | 68.1 | 38.2 | 48.3 |
| | | Summarization | 81.5 | **31.5** | 37.6 | 62.7 | 53.8 | **88.3** | 69.8 | 38.0 | 57.9 |
| | | Clip | **82.5** | 28.5 | 43.6 | **70.0** | 55.9 | 84.1 | 68.7 | **40.5** | **59.2** |
| | CPL | Window | 79.0 | 25.3 | **44.9** | 72.0 | 55.4 | 86.5 | 67.8 | 38.9 | 58.7 |
| | | Long | 46.5 | 18.8 | 32.3 | 46.5 | 52.4 | 78.5 | **69.7** | 38.8 | 47.9 |
| | | Summarization | 77.2 | **31.5** | 39.5 | 65.0 | 56.5 | **88.6** | 68.0 | 40.4 | 58.3 |
| | | Clip | **82.3** | 31.5 | 43.8 | **70.0** | 56.8 | 86.3 | 68.6 | 39.3 | **59.8** +1.2 |
| GPT-4o-mini | / | Window | **82.0** | 52.5 | 37.8 | 55.0 | 89.4 | 97.3 | 73.4 | 40.2 | 66.0 |
| | | Long | 62.0 | **62.0** | **45.5** | 52.0 | 87.2 | 96.8 | 72.9 | 38.4 | 64.6 |
| | | Summarization | 71.0 | 56.2 | 42.2 | 66.7 | 91.2 | **98.9** | 72.9 | 40.7 | 67.5 |
| | | Clip | 81.0 | 54.5 | 39.8 | **81.8** | 92.0 | 98.9 | 75.5 | 45.3 | **71.1** +5.1 |

**Long Context** method preserves the complete conversation history $\mathcal{C}_t$ at each turn, allowing unrestricted access to all previous rounds without using any masking.

**Window Context** method maintains only the most recent $W$ rounds in $\mathcal{C}_t$, masking older interaction rounds.

**Clip Context** method is conceptually defined as periodic attention masking: when the clearing threshold $H$ is reached, the attention mask is reset to attend only to the $L$ most recent rounds, while masking all earlier rounds.

To further optimize computational efficiency, we implement the attention masking scheme through an equivalent context trimming approach in practice. Instead of maintaining full context with selective masking (which zeros out attention to masked rounds), we directly remove those rounds from the input, producing identical model outputs. Taking Clip context as an example:

$$\mathcal{C}_t^{\text{trimmed}} = \begin{cases} \{r_{t-L+1}, \dots, r_t\} & \text{if } |\mathcal{C}_{t-1}^{\text{trimmed}}| + 1 = H \\ \mathcal{C}_{t-1}^{\text{trimmed}} \cup \{r_t\} & \text{otherwise} \end{cases}$$
(2)

**Summary Context** method extends Clip Context by incorporating summarization. When the clearing threshold is reached, a summary model compresses the entire context into a compact summary, which is appended to a summary list. The context is then cleared to retain only the $L$ most recent rounds. At each step, the full summary list and recent rounds are concatenated as input for action generation.

Our clip context management offers three key advantages:

(1) **Balances exploration and exploitation**: Our method periodically alternates between short and long contexts. Short contexts reduce inertia and enable exploration of new strategies, while long contexts leverage interaction history for exploitation. (2) **Improves computational efficiency**: Unlike sliding window approaches that cannot leverage prefix caching, our method is compatible with KV cache, reducing inference overhead. Clip context achieves approximately $W\times$ speedup in prefill computation, where $W$ is the window size. Detailed analysis of KV cache compatibility and computational complexity are provided in Appendices E and F, respectively. (3) **Breaks inertia and prevents error accumulation**: Periodic context clearing to a lower level than window context achieves better error accumulation prevention and lower inertia on average. We further quantitatively demonstrate this inertia reduction through diagonal attention ratio analysis in Section 3.4. We provide a case study in Maze environment to visualize this error accumulation prevention effect in Appendix C.

## 3. Experiments

### 3.1. Environments

We evaluate our approach across eight environments, covering embodied AI (Shridhar et al., 2021), web interaction (Yao et al., 2022), reasoning-dependent strategic games (Guertler et al., 2025), navigation tasks (Abdulhai et al., 2025), and crafting scenarios (Prasad et al., 2024). We use AgentGym (Xi et al., 2024) suite implementation. We also evaluate in Deep Research scenarios based on BrowseComp (Wei et al., 2025), using ReSum (Wu et al., 2025)

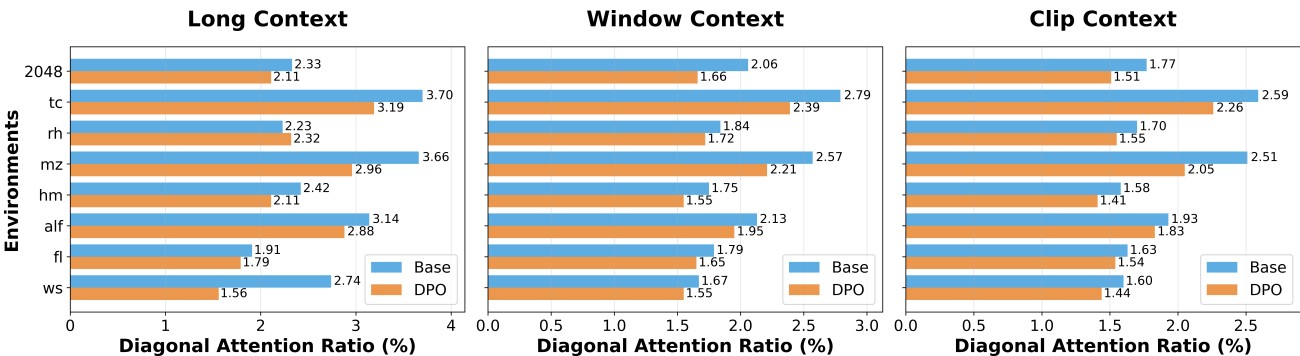

*Figure 5.* Diagonal Attention ratio analysis across different context configurations. Lower ratios indicate reduced conversational inertia.

codebase. For detailed environment specifications and experimental configurations, refer to Appendix N.

### 3.2. Implementation Details

We evaluate our training-free clip context methods using three language models: Qwen3-8B (w/o thinking) (Yang et al., 2025), Llama3.1-8B-Instruct (AI @ Meta, 2024), and GPT-4o-mini. We set temperature to 0.8 for all methods to make the baseline stronger and the comparison more conservative. While standard benchmarks often use temperature=0 for reproducibility in single-turn tasks, multi-turn agent environments benefit from stochastic sampling: higher temperature reduces brittle failure modes caused by deterministic repetition loops and better reflects the exploration–exploitation dynamics inherent in sequential decision-making. For fair comparison, we tune and report optimal hyperparameters for each context management method: Clip-12to1, Window-6, and Sum-12to1. For the Long context baseline, we retain all previous interaction rounds. Summarization implementation details are provided in Appendix I.

### 3.3. Main Results

We evaluate how different context management strategies balance inertia control against information loss.

Table 1 reveals three key findings: (1) **Context Preference Learning performs well in multiple context management**. The CPL-trained models show substantial gains, with Qwen-CPL achieving 72.5% under Clip-context compared to the base model's 68.9%, and similar improvements observed across Window-context, Long-context. (2) **Clip-context outperforms both Window-context and Long-context across all tested models.** Averaging across eight environments, the information gains from longer context are substantially outweighed by their negative impact on performance. (3) **Clip-context achieves comparable performance to summarization-based methods.** We find that reduced inertia from clipping context plays a major role in

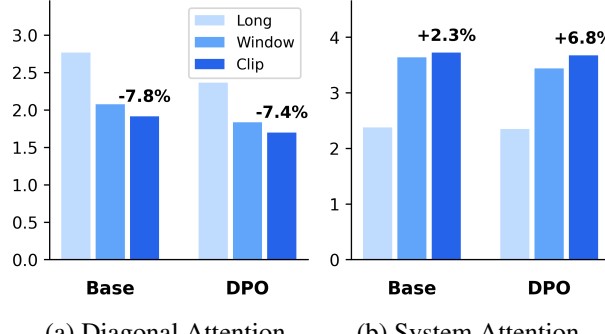

|                  |                     |
|:----------------:|:-------------------:|
| (a) Diagonal Attention | (b) System Attention |

*Figure 6.* Impact of clip context method on attention patterns across eight environments

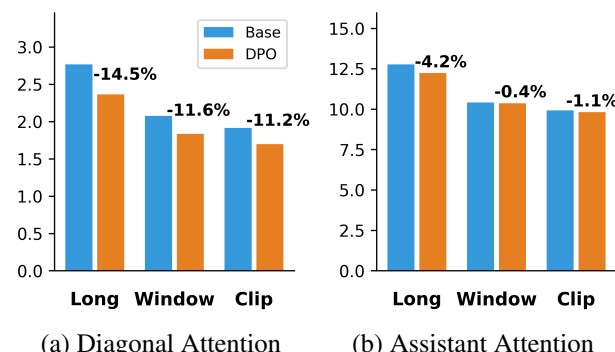

|                  |                     |
|:----------------:|:-------------------:|
| (a) Diagonal Attention | (b) Assistant Attention |

*Figure 7.* Impact of Context Preference Learning on attention patterns across eight environments

summarization methods' performance gains, rather than the generated summarization content itself.

### 3.4. Attention Pattern Analysis

To investigate how clip context methods and Context Preference Learning influence models at the attention level, we design attention-ratio indicators that provide quantitative insights into attention pattern changes.

We extract attention patterns by performing a single forward

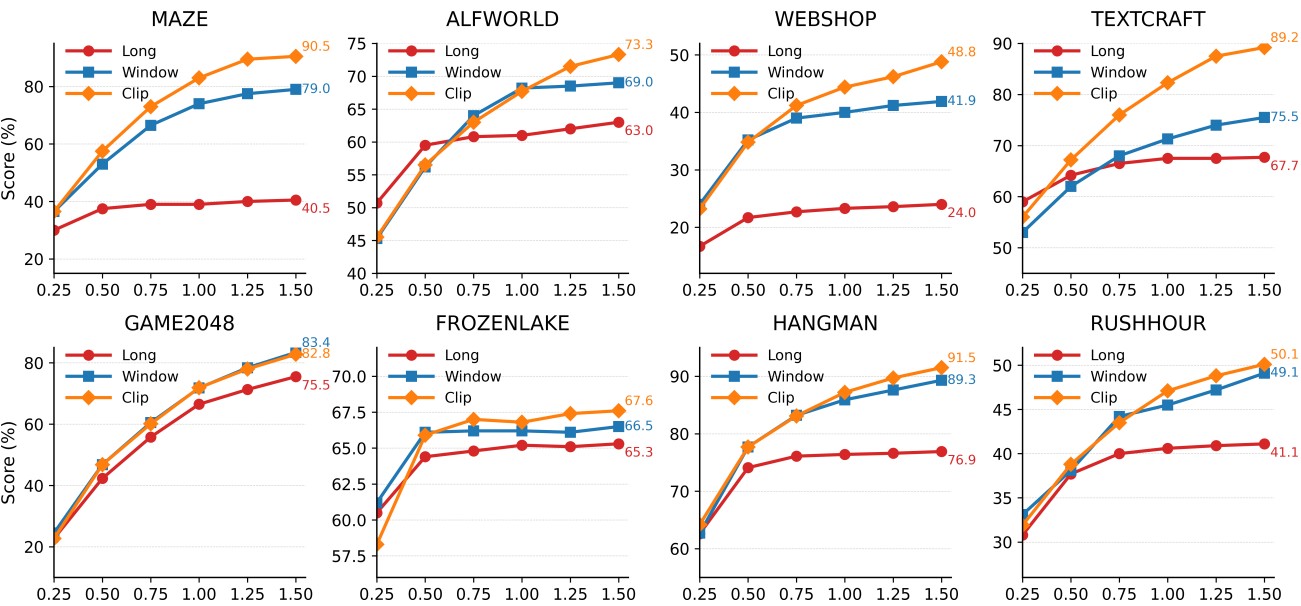

*Figure 8.* Score vs. maximum step scaling analysis. The x-axis represents the ratio relative to the baseline maximum step number.

*Table 2.* Comparison of context management methods on the deep research task (BrowseComp ([Wei et al., 2025](#)) 128x2 case). Proactive rate indicates the percentage of agents that answer before reaching step limits. Proactive score represents the average score when agents confidently provide answers early.

| Method | Score ($\pm$ SEM) | Proac. rate | Proac. Score |
|---|---|---|---|
| Window-6 | $25.0 \pm 1.7$ | 27.3% | 55.7 |
| Window-9 | $23.4 \pm 1.5$ | 26.6% | 58.8 |
| Window-12 | $24.2 \pm 1.8$ | 25.4% | 58.5 |
| Clip-12to0 | $25.0 \pm 1.8$ | 29.3% | 56.0 |
| Sum-12to0 | $27.7 \pm 2.0$ | 63.7% | 37.4 |
| **Clip-12to6** | $\mathbf{29.3} \pm 1.5$ | 30.9% | 61.0 |
| Sum-12to6 | $28.1 \pm 1.6$ | 30.8% | 63.3 |

pass through the model and retrieving the attention matrix from the final layer. We empirically find that the trend of diagonal attention in the final layer is similar to other layers. All attention heads are averaged to create a unified attention matrix. We categorize tokens based on their positional roles: sink tokens (first 3 tokens), system tokens, user tokens, previous assistant tokens, and current assistant tokens. For each output token, we calculate the sum of attention weights directed toward each token category, then average these sums across all output tokens to obtain the final attention ratios for each category. Additionally, we introduce a diagonal-attention indicator that measures conversational inertia. This metric computes attention between the current output and previous assistant responses that occupy corresponding diagonal positions with an expansion of $r = 5$ tokens around the diagonal.

Figure 6 shows that the clip context method reduces diagonal attention by 7% for both base and CPL models, decreasing attention to previous corresponding tokens, which is consistent with mitigating conversational inertia. Simultaneously, it increases system attention, suggesting greater focus on the specific task requirements rather than over-mimicking self-generated content from previous responses.

Figure 7 demonstrates that Context Preference Learning significantly reduces diagonal attention by over 10% across all context management methods. Critically, diagonal attention decreases 10% more than the overall attention to previous assistant responses. Since our preference pair construction treats diagonal and non-diagonal positions identically, this differential effect reduces the plausibility of a trivial reallocation explanation—such as uniformly downweighting all history or simply shifting attention toward recent tokens. Instead, the model selectively reduces diagonal attention beyond its reduction of general assistant history, which would not occur under uniform reallocation. This differential pattern provides targeted mechanistic evidence that CPL learns structural patterns associated with failure, rather than merely redistributing attention across the context.

Furthermore, we observe that the Context Preference Learning exhibits stronger reduction effects on Long context configurations. Across the eight environments, diagonal attention decreases by 14.5% for Long context compared to 11% for the shorter Window and Clip methods. This is because Long context management suffers from the highest inertia, demonstrating that CPL's ability to mitigate inertia becomes stronger when the baseline inertia is higher, effectively bringing inertia back to comparable levels.

*Table 3.* Ablation studies on hyperparameters. All results show 8 environment average scores.

| Hyperparameter | Score |
|---|---|
| L=1, H=2 | 62.2 |
| L=1, H=3 | 66.4 |
| L=1, H=6 | 68.7 |
| L=1, H=12 | 68.9 |

(a) Effect of H parameter

| Hyperparameter | Score |
|---|---|
| L=1, H=12 | 68.9 |
| L=3, H=12 | 67.1 |
| L=6, H=12 | 62.1 |
| L=11, H=12 | 60.7 |

(b) Effect of L parameter

| Hyperparameter | Clip | Window |
|---|---|---|
| L+H=7, W=3 | 68.7 | 64.0 |
| L+H=11, W=5 | 68.8 | 64.2 |
| L+H=13, W=6 | 68.9 | 64.9 |
| L+H=15, W=7 | 68.9 | 63.8 |

(c) Clip vs Window at different lengths

*Table 4.* General capability preservation of CPL-trained models. GPQA-Diamond scores are averaged over 10 runs. We follow the same sampling parameter in Qwen3 (Yang et al., 2025).

| Model | GPQA-Diamond | MMLU-Redux |
|---|---|---|
| *Non-thinking* | | |
| Qwen3-8B | 48.83 | 79.13 |
| Qwen3-8B-CPL | 49.09 | 79.32 |
| *Thinking* | | |
| Qwen3-8B | 58.03 | 83.04 |
| Qwen3-8B-CPL | 57.92 | 83.27 |

*Table 5.* Ablation study on context preference pair construction (Qwen3-8B base, Clip-12to1 evaluation). Chosen and Rejected indicate the number of conversation rounds used to generate actions.

| Chosen | Rejected | 8 env avg score |
|---|---|---|
| - | - | 68.9 (Base) |
| 1 round | 6 rounds | 70.3 |
| 6 rounds | As long as possible | 72.5 |

### 3.5. Clip context in long-term reasoning tasks

**Long-term Reasoning Tasks.** To further evaluate the effectiveness of Clip-context in long-term reasoning scenarios that require deep research and multi-step thinking, which pose higher demands on balancing exploration with information retention, we conduct experiments on BrowseComp. Implementation details are provided in Appendix J.

Table 2 reveals three key findings: (1) **Clip-12to6 outperforms Window-context despite having equivalent min, max, and average input length.** Clip-12to6 achieves better performance than Window-6, Window-12, and Window-9. This demonstrates Clip-context's ability to retain moderate context for state identification while managing inertia dynamically. (2) **Sum-12to0 produces more proactive answers with substantially reduced accuracy.** Sum-12to0 generates earlier responses more frequently, yet accuracy drops significantly (from 56% to 37.4%). We hypothesize this occurs when clipping all interaction history leaves only summarization and current observation, the actor model may be misguided by errors in the generated summary. This phenomenon can be suppressed when provided with several turns of objective interaction history. (3) **Clip-12to6 achieves comparable performance to Sum-12to6.** LLM-generated summarization provides little benefit when objective interaction history is available. More detailed case study analysis is in Appendix K.

### 3.6. Preservation of General Capabilities After Context Preference Learning

We evaluate our Context Preference Learning trained models on standard benchmarks measuring general knowledge and reasoning capabilities in Table 4. The results demonstrate that Context Preference Learning preserves the model's

original capabilities remarkably well.

### 3.7. Long-Horizon Scaling Analysis

To demonstrate the scalability advantages of our approach, we analyze performance across varying episode lengths from 0.25x to 1.5x standard evaluation limits, as shown in Figure 8. The scaling analysis reveals that our clip context outperforms window approaches in long episode settings by breaking inertia through periodic context clearing.

### 3.8. Ablation Study on Preference Pair Construction

To validate the effectiveness of Context Preference Learning, we conduct ablation studies on different preference pair construction strategies. Table 5 shows that preferring moderate contexts (6 rounds) over excessively long ones outperforms preferring short contexts (1 round) over moderate context (6 rounds). This validates our hypothesis that mitigating the negative effects of excessive inertia in overly long contexts is more effective than simply maximizing exploration through minimal context.

### 3.9. Hyperparameter Analysis of Clip Context

To understand the impact of clip context hyperparameters and provide guidance for practical application, we conduct comprehensive ablation studies varying both the clearing interval H and retention length L.

**Effect of H parameter.** The upper limit H provides more contextual information for decision-making, making the agent more informed when proactively answering. Table 3(a) shows the effect when L=1:

**Effect of L parameter.** The lower limit L controls the ability to break inertia: lower L values strengthen the ability to propose new methods and paths, making it more suitable for exploratory tasks; moderate L values retain more informa-

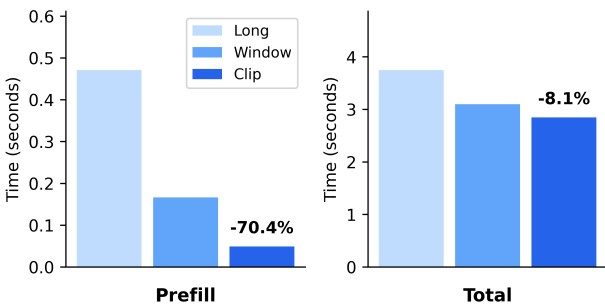

*Figure 9.* Computational efficiency comparison between context management strategy

tion, making it more suitable for summarization and search tasks. Table 3(b) shows the effect when we mainly adjust L:

**Clip vs Window.** When Clip and Window method have the same average input to model, Clip (L=1) outperforms Window at various context lengths (Table 3(c)):

### 3.10. Computational Efficiency Analysis

To validate our clip method speedup by natural KV cache friendly advantages, we conduct comprehensive speed benchmarking across eight environments. As shown in Table 9, the experimental results validate our theoretical analysis: our clip context method achieves 2-7x prefill speedup over window methods across all environments while maintaining stable performance, whereas long context suffers speed degradation in complex environments.

### 4. Related Works

Recent advances in large language models (LLMs) have made In-Context Learning a central topic in NLP. Brown et al. (2020) showed that GPT-3 can adapt to new tasks from a few examples, but later studies found its performance highly sensitive to prompt design (Lu et al., 2022; Webson & Pavlick, 2022; Sclar et al., 2024), indicating that ICL relies on surface-level pattern imitation rather than explicit semantic learning (Min et al., 2022). At the mechanistic level, induction heads in Transformers have been shown to implement copying behavior by attending to prefix patterns and replicating them, thereby offering a plausible basis for ICL (Olsson et al., 2022; D'Angelo et al., 2025). Along this direction, Halawi et al. (2024) identified two failure modes, namely overthinking and false induction heads. Nevertheless, these insights face challenges in long-context scenarios where LLMs exhibit positional bias (Mikhail et al., 2025; Hsieh et al., 2024b) and multi-turn performance degradation (Hong et al., 2025). Laban et al. (2025) found that premature answer attempts and verbosity cause degradation, discussed in Appendix G. Related work has examined context effects in task-switching scenarios, where switching

between different tasks within a conversation leads to performance degradation and task interference (Hankache et al., 2025; Gupta et al., 2024; Castillo-Bolado et al., 2024).

As LLM capabilities have expanded, agentic applications have emerged as a promising frontier. Most prominently, the ReAct framework (Yao et al., 2023b) integrates reasoning with action through alternating reasoning–action paradigms, while enhanced base agents improve decision-making via self-prompting and state-tracking (Rozanov & Rei, 2025). Complementary strategies introduce exploration–exploitation mechanisms through multi-path sampling (Wang et al., 2023) and systematic search approaches (Zhou et al., 2024; Yao et al., 2023a). Planning-based approaches have shown promise through explicit planning separation (Erdogan et al., 2025) and look-ahead strategies (Nikhil Verma, 2025). Training methodologies have evolved with synthetic self-reflected trajectories (Chen et al., 2025) and context summarization for long-horizon tasks (Wu et al., 2025), though challenges persist with identity drift (Choi et al., 2025) and sophisticated reasoning requirements (Zheng et al., 2025; Zhang et al., 2025a;b). Multi-agent orchestrator-based systems have explored collaborative approaches through emergent behaviors (Chen et al., 2024), adaptive team building (Song et al., 2025), and optimizable graph structures (Zhuge et al., 2024), though these require environment-specific architectures and task decomposition capabilities. However, these approaches rely on complex pipeline designs that only indirectly alleviate conversational inertia. They do not address the fundamental nature of long-horizon agent problems and lack explicit mechanisms to tackle error propagation caused by induction heads.

Context management methods for long-horizon agents broadly fall into three categories: sliding-window approaches (Xiao et al., 2024) that maintain a fixed context budget, summarization-based methods (Wu et al., 2025) that replace raw history with compressed summaries, and retrieval-based methods that augment context with retrieved past experience. All three treat performance degradation as a consequence of context length or information overload, without isolating the role of self-imitation bias in the model's own outputs. Our work identifies conversational inertia as a distinct failure mode arising from this bias, and proposes methods that target this mechanism directly. Clip Context and CPL are orthogonal to and combinable with existing context management strategies, as demonstrated in Appendix P.

### 5. Discussion

While we designed clip context primarily for short-term tasks such as gaming environments, it does not directly address information loss in long-horizon tasks. However,

empirically we observe performance improvements even in long-term tasks such as deep research, attributable to its inertia mitigation effects. Clip context achieves comparable performance to LLM-based summarization while avoiding additional computational overhead. How to integrate low inertia while preserving task-relevant information remains an open question.

**Limitations.** We acknowledge three key limitations of the current work. First, Clip Context inherently involves an information-inertia tradeoff: periodically clearing context can cause failures in tasks with strong long-range dependencies, particularly when critical early information falls outside the clipped window (see Appendix L for failure case analysis). Second, while moderate defaults ($L{=}1{-}3$, $H{=}2W$) generalize well, optimal Clip parameters can vary across environments and may benefit from task-specific tuning (see Appendix M). Third, our analysis focuses on conversational inertia as one contributor to multi-turn degradation; other factors such as error propagation and reward sparsity are outside the scope of this work.

## 6. Conclusion

This work identified conversational inertia as a previously unrecognized contributor to multi-turn agent performance limitations. Our analysis reveals that diagonal attention patterns to previous responses are associated with constrained exploration capabilities. The proposed framework, combining Context Preference Learning and clip context, demonstrates effectiveness in mitigating these patterns. Experimental results indicate that our framework improves agent performance while reducing computational overhead, suggesting this phenomenon merits further investigation in multi-turn system design.

## Reproducibility statement

To ensure reproducibility of our results, we provide comprehensive implementation details throughout the paper and appendices. Section 3.2 specifies the exact models used (Qwen3-8B, Llama3.1-8B-Instruct, GPT-4o-mini), evaluation temperature (0.8), and hyperparameters for both clip context (H=12, L=1) and window context (W=6) methods. Our Context Preference Learning approach uses LoRA fine-tuning with detailed configurations in Appendix D.1, including rank=16, alpha=16, learning rate=5e-7, and dataset collection procedures with K=20 step minimum context gaps across 1,000 preference pairs per environment. Environment specifications are provided in Appendix N with exact step limits, reward structures, and evaluation protocols for all eight environments using the AgentGym framework. All attention analysis procedures are specified in Section 3.4 with r=5 token diagonal expansion and final-layer attention extraction methods.

## Generative AI Considerations

The authors confirm full responsibility for all content in this paper. We used Large Language Models (LLMs) to assist in the following ways: (1) to search related works and identify relevant literature connections across the field of multi-turn dialogue agents and conversational inertia; (2) to aid and polish writing, including spell checking, finding appropriate vocabulary to express intended meanings, and generating initial drafts of paragraphs that were subsequently revised and validated by the authors. All core research content, methodology, experimental design, results, and technical contributions represent the original work of the authors. All LLM-generated content has been carefully reviewed, edited, and verified by the authors to ensure accuracy and scientific rigor.

## Ethical Conduct for Peer Review

The authors confirm adherence to standard ethical conduct for peer review. This work has not been advertised as under submission to ICML during the review period through talks, social media, or other public channels. The authors have not engaged in any form of collusion with reviewers, area chairs, or senior area chairs. All content in this paper represents original work by the authors, with proper citations to prior work. The authors confirm that no prompt injection techniques have been employed in this manuscript.

## Impact Statement

**Ethical Considerations.** We believe that our proposed conversational inertia mitigation framework raises no ethical concerns regarding its motivation, design, implementation,

or data usage. The method is designed to advance multi-turn agent systems in a principled and efficient manner by addressing the challenge of imitation bias that constrains exploration capabilities. Our Context Preference Learning approach requires no environment-provided rewards or expert demonstrations, enabling responsible development without reliance on potentially biased supervision signals. The framework adheres to ethical guidelines in AI research while promoting transparent understanding of attention mechanisms in language models.

**Societal Implications.** Our framework introduces a new perspective in multi-turn agent design by identifying and mitigating conversational inertia, where models erroneously mimic their own previous responses as few-shot examples. By addressing this limitation through model-level preference calibration and inference-time context control, our approach enables more reliable autonomous agents across diverse applications including web navigation, embodied AI, and interactive task completion. The combination of improved agent performance and reduced computational costs through efficient context management has the potential to democratize access to intelligent automation systems. Furthermore, our analysis of diagonal attention patterns provides insights that can inform future architectural designs for multi-turn systems, advancing the broader understanding of how language models process conversational history in sequential decision-making scenarios.

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

# A. Transfer of Few-Shot Learning to Agent Domains

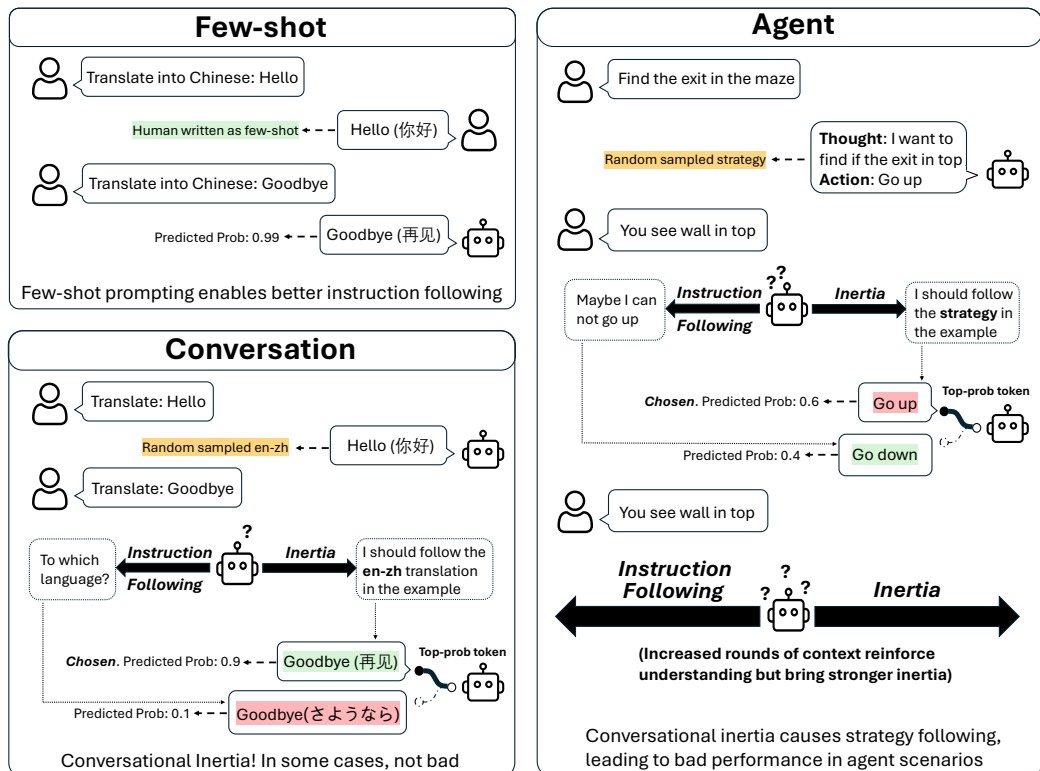

*Figure 10.* Illustration of LLMs incorrectly transferring few-shot capabilities to agent domains. In standard few-shot learning scenarios, models learn from high-quality demonstrations. However, in multi-turn agent interactions, models inappropriately apply this learning pattern to their own previous responses, creating conversational inertia that degrades performance.

# B. Attention Distribution Analysis Details

This section presents additional attention ratio data and discussion, as complementing Section 3.4.

As shown in Figure 2, different attention component ratios exhibit distinct trends as the number of context rounds increase. We hypothesize that diagonal attention in previous assistant responses primarily plays a harmful role in behavioral imitation. However, assistant attention, user attention, and system attention should be maintained within moderate ranges to balance exploration-exploitation. Although excluding any history from context can increase exploration by making system attention much stronger and user/assistant attention lower, this approach misses important contextual information needed to better exploit interaction history, resulting in significantly low performance.

The experimental results in Figure 11 show two key findings: (1) Context Preference Learning significantly reduces diagonal attention while maintaining minimal changes to overall assistant attention. This successfully redirects diagonal attention to other parts of the assistant's processing, enhancing holistic understanding rather than positional imitation, while keeping user and system attention relatively stable. (2) Context clipping reduces attention to both assistant and diagonal components while slightly increasing attention to system and user inputs. By directly controlling context, this approach focuses more on the balance of exploration-exploitation rather than learning from self-generated few-shot examples of unreliable quality.

# C. Context Influence and Breaking inertia analysis in Maze

To investigate how initial context influences subsequent model behavior and evaluate the effectiveness of different context management methods in breaking conversational inertia, we conduct a controlled experiment in the maze environment. As shown in Figure 12, we fix the starting state (position and task) while designing two different initial contexts. The two context types are: (1) a *good init example* containing an optimal trajectory (left, left, up, up) that moves directly toward the

current position, and (2) a *bad init example* featuring a suboptimal looping pattern (right, left, right, left) that wastes steps. This design aims to verify whether agents follow initial context examples and whether inertia phenomena exist that lead to performance differences.

Our experimental findings reveal two key insights that validate the existence of conversational inertia and demonstrate our method's effectiveness. (1) **Strong Context Following Behavior**: Models exhibit pronounced sensitivity to initial context, with bad initial examples leading to suboptimal exploration patterns across all methods. This empirically confirms that conversational inertia significantly impacts agent behavior in multi-turn scenarios shown in Figure 10. (2) **Clip Context Effectiveness**: Our clip context method successfully mitigates the negative impact of poor initial context while maintaining effective exploration strategies. Unlike full context and window context methods that remain heavily influenced by suboptimal initial examples, clip context demonstrates robust performance regardless of initial context quality.

We fix the starting state (position and task) while designing two different initial contexts. The two context types are: (1) a *good init example* containing an optimal trajectory (left, left, up, up) that moves directly toward the current position, and (2) a *bad init example* featuring a suboptimal looping pattern (right, left, right, left) that wastes steps. We execute 64 episodes with a 60-step limit from identical starting states. We visualize position visit frequencies as heatmaps to analyze exploration patterns and step efficiency across context management methods, where visit percentage represents the proportion of total steps spent at each position. Window context uses $W = 6$, while Clip context maintains $H = 12, L = 1$.

As shown in Figure 12, our experimental findings reveal several key insights:

**Strong Context Following Behavior:** Comparing the first row (good init example) with the second row (bad init example) across all three columns shows dramatic performance differences. We control for identical starting states, action spaces, and optimal paths. However, performance varies drastically between good and bad init examples. This demonstrates that models incorrectly transfer strong few-shot learning capabilities, automatically treating previous agent actions in context as examples to imitate, regardless of their quality. This finding aligns with our observations in Figure 10, confirming that models can significantly degrade performance when exposed to patterns that are implicitly accepted without scrutiny.

**Persistent Context Influence and Window Context Limitations:** Examining the Window context column (middle column) reveals that the quality of the initial 4-step context profoundly affects subsequent 60-step performance across all context management methods. Even though early history is quickly dropped in both Window context and Clip context approaches, the influence of good versus bad init examples persists throughout episodes, significantly impacting final scores. This demonstrates the lasting effect of early context on model behavior and shows that simply limiting recent history cannot break error loops. While Window context provides modest improvements under good init examples compared to Full context, it fails to improve performance under bad init examples, highlighting its inability to escape negative patterns. Although Window context removes outdated steps and excludes the initial 4-step bad example, it consistently maintains the most recent 6-step context, causing negative influences to persist throughout episodes and resulting in extremely low completion rates.

**Clip Context Effectiveness:** Comparing the third column (Clip context) with the first two columns shows that only Clip context successfully mitigates the impact of bad init examples. Most notably, Clip context achieves 32.8 score under bad init examples compared to Window context's 3.1, demonstrating a 10.6× improvement in breaking free from negative patterns. Analysis of visit percentage distributions reveals that Clip context shows fewer instances of getting stuck at individual positions compared to Full context or Window context methods. In the optimal direction (5-th row, rightmost position), Clip context achieves 12% visit frequency under good init examples and 8.3% under bad init examples, higher than Window context's 5.1% and 1.6% respectively. Conversely, in suboptimal directions (upper right regions moving away from the goal), Clip context shows lower visit frequencies than Window context. Furthermore, under good init examples, Clip context's highest single-position visit percentage is 19%, lower than Full context's 21% and Window context's 24%, indicating more balanced strategic exploration.

## D. Additional Training Experimental Details

### D.1. Training Configuration

We implement DPO training using LoRA (Low-Rank Adaptation) fine-tuning, which approximates weight updates through low-rank matrices to achieve parameter-efficient training. For a weight matrix $W \in \mathbb{R}^{d \times k}$, LoRA decomposes the update as:

$$W' = W + \frac{\alpha}{r} \cdot A \cdot B \tag{3}$$

where $A \in \mathbb{R}^{d \times r}$ and $B \in \mathbb{R}^{r \times k}$ are trainable low-rank matrices. The rank $r$ determines the dimensionality of the low-rank decomposition (controlling model expressiveness), while alpha $\alpha$ controls the scaling factor for the adapted weights (regulating adaptation strength). Our main training configuration employs LoRA fine-tuning with rank 16 and alpha 16, targeting 0.4% of model parameters. We use DPO loss with beta=0.01, and sigmoid loss type. Training runs for 2 epochs with learning rate 5e-7 using AdamW optimizer.

Our DPO data collection uses a minimum context gap of $K = 20$ steps to ensure sufficient disparity between long and short context windows. We collected 1,000 preference pairs per environment, yielding an 8K total dataset across all environments.

Hardware setup includes 4×A100 GPUs for training and attention visualizing, 8×RTX4090 for evaluation, and 1×A6000 for computational efficiency benchmarking.

### D.2. LoRA Ablation Study

To investigate whether the model learns environment-specific patterns or develops generalizable capabilities, we conduct DPO training using an Ultra-Low Parameter (Rank=1) experiment. By severely constraining the trainable parameter count to be far smaller than the dataset size, this setup prevents environment overfitting and forces the model to learn unified capabilities across environments. We use the same eight environment dataset as the main experiments but with extremely constrained parameters. We employ LoRA rank=1 with alpha=32, resulting in a high alpha/rank ratio of 32. To ensure training stability with this aggressive parameter reduction, we implement gradient accumulation steps of 4 and DPO label smoothing of 0.1. This configuration trains only 0.02% of the model parameters while maintaining effective learning dynamics. All other training parameters remain identical to the main DPO configuration.

Besides, to understand the impact of hyperparameter choices on our Context Preference Learning approach, we conduct ablation studies on LoRA alpha values while maintaining rank=16. We evaluate three configurations: alpha=12, alpha=16, and alpha=24 across all eight environments using Qwen3-8B as the base model.

*Table 6.* Ablation study of LoRA training parameters for Context Preference Learning. Success rates (%) are reported for different parameters. Evaluation with Clip Context Strategy

| Rank | Alpha | MZ | ALF | WS | TC | FL | HM | 2048 | RH | Avg |
|---|---|---|---|---|---|---|---|---|---|---|
| 1 | 16 | 89.0 | 67.0 | 48.7 | **86.0** | **71.6** | 85.6 | 72.4 | 48.0 | 71.0 |
| 16 | 12 | 86.0 | **73.0** | 50.7 | 84.0 | 69.4 | 83.6 | **74.0** | 49.1 | 71.2 |
| 16 | 16 | **91.5** | 70.3 | **54.9** | 83.0 | 68.4 | 87.5 | 73.0 | 51.1 | **72.5** |
| 16 | 24 | 91.0 | 70.5 | 51.2 | 79.5 | 67.2 | **88.8** | 72.9 | **52.3** | 71.7 |

The ablation results reveal that alpha=16 provides the optimal balance across most environments, achieving the highest average performance of 72.3%. While alpha=24 shows competitive performance in certain environments (ALF, HM, RH), it exhibits reduced performance in strategic reasoning tasks (TC, WS, 2048). This suggests that moderate regularization strength (alpha=16) better preserves the model's original capabilities while effectively incorporating the conversational inertia mitigation preferences.

## E. Continuous Cache Pruning vs Discrete History Truncation

This section first describes and categorizes the work on KV Cache optimization, then explains why Window Context cannot use KV Cache Reuse.

A large body of work on attention optimization for LLMs can be broadly grouped into two paradigms: continuous pruning and discrete pruning. The distinction lies not in the granularity of tokens versus rounds, but in whether pruning is performed in a continuous or blockwise fashion.

Continuous pruning methods operate in a fine-grained, token-continuous manner. Representative approaches include sliding-window attention, KV-cache eviction, and other forms of local or structured sparsity. These methods prune context progressively, allowing partial information flow to be preserved across adjacent segments. Their theoretical complexity is low and they are widely used in long-context modeling.

Discrete pruning methods operate in a coarse-grained, blockwise manner. Examples include prefix caching, history clearing, and round-level memory selection in multi-turn agent settings. Instead of preserving continuity, these methods explicitly remove or reset entire chunks of context, which often aligns better with the online nature of interactive agents.

Remark on the Window Context method. Although the sliding-window attention mechanism is often discussed under long-context modeling, in the present work we treat it as a discrete-level pruning strategy as nearly all agent works do. This is because each window defines a hard cutoff beyond which past information cannot be reused, preventing effective KV-cache reuse across multiple dialogue turns. If we forcibly apply KV cache reuse in Window Context, the results would theoretically differ from the non-optimized Window Context version, effectively becoming an approximation of Full Context with stronger inertia effects. As a result, Window Context cannot utilize KV Cache between rounds theoretically and is generally inefficient for agent-style tasks.

## F. Computational Complexity Analysis

We analyze the computational advantages of our approach through prefill cost comparison:

*Window Context Complexity:* With window size $W$, each inference step processes a context of length $W$ rounds. Since the oldest round is dropped at each step, prefix caching cannot be utilized due to constantly changing context boundaries. The prefill cost per step is $O(W^2)$ attention operations.

*Clip Context Complexity:* Our method alternates between context lengths $L, L+1, \ldots, H-1$ within each clearing cycle. Crucially, all contexts within a cycle share the same prefix up to the last clearing point, enabling effective KV cache utilization. Specifically, when processing context length $i+1$, the KV cache from the previous context length $i$ can be reused, requiring only incremental computation for the new round. This means only the marginal cost $O(i)$ (instead of $O(i^2)$) is needed for each step after the initial prefill. The average prefill cost becomes:

$$\text{Average Cost} = \frac{O(L^2) + \sum_{i=L+1}^{H-1} O(i)}{H - L} = \frac{O(L^2) + O((H-1)^2 - L^2)/2)}{H - L} \tag{4}$$

To compare with sliding window methods fairly, we consider the case when $L = 1$ and $H = 2W$ (equivalent total context budget, maintaining the same average input rounds for fair computational comparison). The speedup ratio becomes:

$$\text{Speedup} = \frac{\text{Window Cost}}{\text{Clip Cost}} = \frac{W^2}{\frac{1 + ((2W-1)^2 - 1)/2}{2W - 1}} = \frac{2W^2(2W-1)}{2 + (2W-1)^2} \approx W \tag{5}$$

This analysis demonstrates that our clip context method achieves approximately $W\times$ speedup in prefill computation compared to sliding window approaches, with the computational advantage stemming from KV cache reuse within each clearing cycle.

## G. Diagonal Attention: Alternative Explanations and Causal Direction

**Disclaimer:** We do not claim definitive causal identification between diagonal attention and performance degradation. Instead, we provide converging mechanistic evidence that supports diagonal attention as a plausible mechanism underlying conversational inertia.

Prior mechanistic work on induction heads and prefix copying (D'Angelo et al., 2025) provides theoretical support for our findings. These studies demonstrate that induction heads implement copying behavior by attending to previous patterns and replicating them, which is consistent with our observed diagonal attention patterns. This mechanistic understanding suggests that diagonal alignment may encode imitation dynamics that accumulate errors across turns, offering a plausible mechanism for *conversational inertia*.

Alternative explanations, such as premature answering and verbosity (Laban et al., 2025), could be interpreted as related symptoms. Our Context Preference Learning selectively suppresses diagonal attention while leaving other attention flows intact, and this targeted reduction consistently correlates with performance improvements (Figure 7). This selective intervention pattern is consistent with diagonal attention being harmful, though we acknowledge this does not constitute definitive causal proof. The evidence points toward diagonal attention as a likely contributor to performance degradation, warranting further investigation through controlled interventions.

## H. Statistical Significance and Variance Analysis

To ensure the reliability of our experimental results, we report standard error of the mean (SEM) across all experiments. Table 7 presents SEM values for all model configurations across eight evaluation environments. The consistently low SEM values validate the statistical significance of our findings.

*Table 7.* Standard error of the mean (SEM) for all experimental configurations across eight environments. All values are reported as percentages. The low SEM values relative to performance differences validate the statistical significance of our findings.

| Model | Context | MZ | ALF | WS | TC | FL | HM | 2048 | RH | Avg |
|-------|---------|------|------|------|------|------|------|------|------|------|
| Qwen3-8B | Window-6 | 3.59 | 1.06 | 0.08 | 0.75 | 1.55 | 1.10 | 1.66 | 3.04 | 0.57 |
| Qwen3-8B | Clip-12to1 | 1.86 | 1.24 | 1.73 | 0.65 | 2.10 | 1.14 | 1.39 | 2.45 | 0.56 |
| Qwen3-8B-CPL | Clip-12to1 | 2.17 | 0.18 | 0.99 | 0.94 | 1.78 | 0.94 | 0.73 | 1.61 | 0.41 |

The uncertainty estimation methodology is as follows: for each task in each dataset, we sample multiple times to obtain multiple average scores per benchmark. We then calculate the standard error across these average scores for each benchmark separately. The overall aggregated uncertainty across all eight environments is computed as the sum of individual standard errors divided by $\sqrt{8}$, providing a normalized measure of cross-benchmark uncertainty.

## I. Summarization Implementation Details

We follow the prompt from ReSum (Wu et al., 2025). For each clip, we pass all context history to generate a summary and append this newly generated summary into a list. Each round when the actor is called, we pass all generated summaries to it.

To prevent a stronger summarization model from directly guiding the actor model and inflating performance, we use the same size model for both summarization and actor roles in our experiments.

For Qwen3-8B and Llama3.1-8B-Instruct, we use them as both the summarizer and actor. For deep research scenarios, as Tongyi Deep Researcher is a specialized model, we use the comparable-sized Qwen3-30B-A3B-Instruct-2507 as the summarizer.

## J. BrowseComp Evaluation Details

We evaluate clip context and summarization methods on BrowseComp (Wei et al., 2025). We use Tongyi Deep Researcher as the actor and Qwen3-30B-A3B-Instruct-2507 as the summarizer. We deliberately choose models of comparable size to ensure the summarizer does not leak information to the actor, which would allow the summary to substitute for decisions that should be made by the actor itself.

To reduce evaluation variance, we first run the same actor for 12 steps across all configurations. Subsequently, all settings continue from this common 12-step baseline, ensuring identical early-stage trajectories and reducing bias. We evaluate on the first 128 cases from BrowseComp, with each configuration run twice.

**Standard Error Calculation:** We report a standard error of the mean (SEM) that reflects **only within-case variability**, avoiding inflation caused by differences in case difficulty. For each evaluation case, we compute its empirical accuracy $\hat{p}_i$ from $k_i$ repeated judgments, and estimate its variance as $\hat{p}_i(1 - \hat{p}_i)/k_i$. Treating cases as fixed and randomness as arising solely from repeated evaluations, the SEM of the overall accuracy is then computed as

$$\text{SEM} = \frac{1}{N}\sqrt{\sum_{i=1}^{N} \frac{\hat{p}_i(1 - \hat{p}_i)}{k_i}},$$

where $N$ is the number of cases. This measures uncertainty due to evaluation noise rather than case difficulty.

**Evaluation Metrics:** We measure three complementary metrics: (1) Overall score based on answer correctness, (2) Proactive answer rate measuring the percentage of cases where agents answer before reaching step limits, indicating confidence in gathered information, and (3) Accuracy split between proactive and forced answers, revealing whether agents can accurately judge when sufficient information has been collected versus being forced to guess at the deadline.

**Summarization Implementation:** We implement the ReSum (Wu et al., 2025) summarization approach using their open-source codebase with minimal modifications to ensure compatibility with our experimental framework.

# K. Empirical Study of Summary-Based Context Management

Existing summarization approaches show unstable performance, suffering from context collapse and brevity bias. Through our empirical study, we also found critical issues with summary-based methods, including over-confidence: model-generated summaries may lead the actor model to have higher proactive answer rates but lower answer accuracy. Therefore, our method does not adopt summarization. In contrast, our Clip method is simple yet effective, and we recommend it as a strong baseline.

## K.1. Quantitative Analysis and Case Study

Our experiments reveal that from a quantitative perspective, summaries encourage the actor model to make decisions even when the prompt does not explicitly require it, while simultaneously reducing decision accuracy. This phenomenon becomes more pronounced when only summary content is retained after clipping, without bare history turn information, as shown in Table 2.

We conducted an empirical case study using Qwen3-30B-A3B-Instruct-2507 as the summary model. From a dataset of 15 failed actor cases, we analyzed whether the summaries contributed to the failures. We identified three major categories comprising four sub-categories of common errors:

**Over-claiming:** Making definitive assertions about ambiguous information.

- Using definitive language for uncertain matters, as demonstrated in Example 1.

**Premature Conclusion:** Drawing conclusions that should be deferred to the actor.

- Making absolute claims based on insufficient evidence, preempting the actor's judgment and ignoring exploration possibilities, illustrated in Example 2.

**Missing Information:** Omitting critical context or findings.

- Omitting critical failed path information, which may lead the actor to repeat verified error directions, exemplified in Example 3.

- Omitting previous actor key findings, which may cause the actor to redundantly search again.

Table 8 presents the distribution of error types as percentages.

*Table 8.* Distribution of Summary Error Types in Failed Cases

| Error Type | Percentage |
|---|---|
| Definitive language for uncertain matters | 66.7% |
| Overly absolute claims | 60.0% |
| Missing critical failed path information | 46.7% |
| Unmentioned key findings | 46.7% |

## K.2. Representative Examples

We present representative examples demonstrating each error category with their problematic summaries highlighted.

---

**Example 1: Over-claiming** (Sample 22)

*Summary excerpt:* "The 1990s short-lived TV series that matches the criteria is 'Savannah,' which aired on ABC from August 23 to September 13, 1996. The actor born in Tennessee is Leslie Jordan, who was born in Memphis, Tennessee, and appeared in the series. The actor who was a Caribbean immigrant is Eva LaRue, who is of Mexican descent; however, no evidence in the conversation confirms she is a Caribbean immigrant. The conversation does not identify a Caribbean immigrant actor in the cast of 'Savannah.'"

*Issue:* The summary opens with a definitive assertion "The 1990s short-lived TV series that matches the criteria is 'Savannah,'" yet immediately contradicts itself by stating that no evidence confirms key criteria. This over-claiming problem uses absolute language like "matches the criteria" while the actual conversation failed to verify all three actor requirements, particularly the Caribbean immigrant and the law enforcement father criteria

---

**Example 2: Premature Conclusion - Absolute Claims** (Sample 5)

*Summary excerpt:* "The repeated mention of 'Bath UK author' across various searchqueries, despite the lack of verifiable content on the cited websites, indicates that Bath is the city associated with the author in the context of the alliterative titled short story published in September 2023."

*Issue:* This represents an unjustified leap from unreliable mentions to a definitive conclusion, preempting the actor's judgment. The summary acknowledges "lack of verifiable content" yet still "indicates" a definite answer, replacing the actor's decision-making role with a premature conclusion based on insufficient evidence.

---

**Example 3: Missing Information - Failed Path** (Sample 43)

*Summary excerpt:* "The conversation history does not provide sufficient information to confirm that Yuri Seung is the specific researcher in question or to determine the number of relevant PubMed publications."

*Issue:* While this acknowledges uncertainty, it fails to explicitly highlight that no researcher identity could be confirmed, which is critical information for the actor's decision-making. The summary omits the failed exploration paths that would help the actor understand why verification was impossible, potentially causing redundant searches.

---

**Example 4: Maze GPT-5-mini - Badly organized** (Sample 7)

*Summary excerpt:* "- Essential Information:
- Goal position: (8, 6).
- Current position progression:
- Started at (1, 9) with a wall above.
- Moved to (1, 8) with walls above and below.
- At (1, 7), walls on left, above, and below; only right was open, leading back to (1, 8).
- Repeated oscillations between (1,9) and (1,8) and occasionally (1,7), indicating a vertical corridor at x=1 with vertical movement blocked at y=8,9,10 and left blocked at (1,7).
- A single right move from (1,9) reached (1,10) where walls were above and below.
- Last known position: (1, 9) with a wall above.
- Constraints observed:
- At (1,8) and (1,10): walls above and below (vertical movement blocked).
- At (1,7): walls on left, above, below (only right allowed).
- At (1,9): wall above; no info about other directions there.
- Navigation outcome so far: No net progress toward goal; trapped cycling near x=1 with vertical blocked; exploration to the right from (1,8)/(1,9) appears necessary to escape."

---

# L. Discussion on information loss

To empirically investigate the trade-off between information gain and inertia, we randomly selected 50 tasks in AlfWorld and evaluated them using the Qwen3-8B base model. Each task was executed 16 independent times under two configurations: Clip-12to1 and Window-6. Clip-12to1 represents longer context retention but potentially stronger inertia, while Window-6 provides less historical information but may enable more adaptive behavior.

We observed substantial performance gaps between the two settings across different task types. Notably, 20% of tasks

exhibited success rate differences of 30% or more. Table 9 shows the distribution of success rate differences (Clip-12to1 minus Window-6) across tasks.

*Table 9.* Distribution of success rate differences between Clip-12to1 and Window-6 configurations in AlfWorld. Positive values indicate Clip-12to1 performs better; negative values indicate Window-6 performs better.

| Difference Range | [-0.4, -0.3) | [-0.3, -0.2) | [-0.2, -0.1) | [-0.1, 0.1) | [0.1, 0.2) | [0.2, 0.3) | [0.3, 0.7) |
|---|---|---|---|---|---|---|---|
| Count | 1 | 1 | 3 | 24 | 10 | 2 | 9 |

Through qualitative analysis of tasks where each configuration excelled, we identified characteristic patterns consistent with the information-inertia trade-off:

**Clip-12to1 advantages:** In case 2541, which requires accessing multiple different interaction types, the sequential inspection task requires a memory of previously checked items and current state to avoid repeated exploration. Similarly, case 2582 involves coherent multi-step sequential operations with strong sequential dependencies. When context length falls below the dependency span, task success becomes nearly impossible. Clip-12to1's extended context enables proper tracking of these long-term dependencies.

**Window-6 advantages:** In case 2531, the agent encounters ambiguous feedback such as "nothing happens," which typically signals operation failure in most environments. However, in AlfWorld, this feedback indicates successful execution without visible effects. A similar phenomenon is observed in case 2425, where the agent using Clip-12to1 encounters an unexpected result and believes it is unsolvable, so it repeatedly outputs an "exit" action until the history is clipped. Based on these case analyses, Clip-12to1 tends to persist with conventional interpretations due to stronger historical influence, ultimately failing to resolve the task. Window-w6, with less historical context and more frequent clipping, exhibits more flexible interpretation patterns and successfully adapts to the environment's unconventional feedback semantics.

This experiment demonstrates that information loss and inertia represent a trade-off, directly limiting the scalability of fixed-limit window-based context management approaches. For contexts within our maximum context round size, our clip method addresses this limitation by maintaining low inertia while providing higher information retention capacity, achieving a more favorable balance. However, tasks requiring context beyond the maximum context round size remain inherently challenging for our approaches.

Despite the above failure cases, we note that compared to Long context (which retains full history without any information loss), Clip-12to1 achieves a higher overall success rate on ALFWorld as shown in Table 1. Additionally, we follow the same case study setting mentioned above and find that in tasks (e.g., Case 2531, 2425) with difficult non-intuitive operations, shorter context methods have less inertia and are less prone to being trapped in loops, thus having a higher probability of exploring correct directions. This demonstrates that while individual tasks may fail due to information loss, the overall performance benefits from reduced inertia.

**Information loss is an unsolved challenge across all context management approaches.** We emphasize that the information loss challenge is inherent to existing context management strategies, not unique to Clip. The Window method drops conversation turns beyond the window boundary. The summarization approach suffers from lossy and biased compression. As shown in Appendix K, summaries frequently exhibit over-claiming, premature conclusions, and missing critical information. Neither window methods nor summarization approaches adequately address the information loss problem. Our Clip method does not aim to address the information loss challenge. Instead, we identify and address a different problem—conversational inertia—which has been overlooked in prior work.

**Inertia reduction and information preservation can be addressed separately.** We agree that in some cases more information is valuable. However, our key insight is that inertia and information access are two different dimensions that can be addressed separately and combined effectively. Table 2 demonstrates this on BrowseComp. We can see that: (1) reducing inertia can outweigh information loss, as Clip-12to1 achieves better performance than Window-12 despite introducing some information loss by discarding history; (2) the information loss drawback can be recovered by combining Clip with summarization methods, where Sum-12to0 performs better than Clip-12to1; (3) alternatively, directly retaining some recent history as context also proves effective, as demonstrated by Clip-12to6 performing better than Clip-12to1.

The above results imply that Clip's inertia reduction can be effectively combined with other information-preserving methods. Besides, the need to reduce inertia exists broadly in long-horizon multi-round agent scenarios: even a RAG-enhanced agent with task-relevant information retrieval still encounters conversational inertia problems when queries form multi-round interactions in its conversation context history. As a promising future direction, this inertia and information loss tradeoff

could be mitigated by combining inertia-aware adaptive dropping with information-based retention, allowing critical earlier turns to be preserved while still reducing inertia.

## M. How to select Clip hyperparameters

Clip parameters can be easily selected based on task characteristics, with lower sensitivity compared to Window methods. Window context is a special case of Clip (Window $W$ is completely equivalent to Clip $L=W$, $H=W+1$). When we decrease $L$ while increasing $H$, the average input information remains constant, but the model's disruption of inertia becomes stronger when refreshing context.

In practice, the $H$ parameter of Clip can be kept at Window $W$ or slightly higher. For the $L$ parameter: in multi-turn scenarios where agent actions cause state transitions (e.g., navigation tasks), setting $L$ to a lower value is sufficient ($L=1$ or $L=3$). For tasks where each step involves equal reasoning (deep research or other multi-step reasoning), $L$ needs to be set to a moderate value ($L=6$).

Adding a new hyperparameter $L$ does not increase the difficulty of practical application. Window context is widely used in agent systems, and existing approaches (e.g., UI-TARS (Qin et al., 2025)) already require hyperparameter tuning based on experience. We believe Clip's dual parameters do not impose greater tuning burden than Window methods.

Table 10 provides a full per-environment breakdown across Clip and Window configurations. Clip consistently shows lower variance across hyperparameter settings than Window, making it a safer default for new environments. At the same average input length, varying the Clip hyperparameters (Clip-10to1, 12to1, 14to1) leads to nearly identical average scores, while Window variants (Window-5, 6, 7) fluctuate more. A conservative default of $L=1$–$3$ with $H=2W$ consistently matches or exceeds the best Window performance without per-environment tuning.

*Table 10.* Per-environment results for Clip and Window hyperparameter configurations (Qwen3-8B). Success rates (%) are reported. Sum-12to1 is included for reference.

| Method | MZ | ALF | WS | TC | FL | HM | 2048 | RH | Avg |
|---|---|---|---|---|---|---|---|---|---|
| Clip-12to1 | 83.0 | 67.7 | 44.4 | 82.3 | 69.7 | 85.1 | 70.9 | 50.2 | 69.2 |
| Clip-12to3 | 73.0 | 69.0 | 45.6 | 82.5 | 66.2 | 84.2 | 73.3 | 43.0 | 67.1 |
| Clip-12to6 | 53.0 | 67.5 | 42.4 | 76.5 | 65.0 | 80.5 | 67.4 | 44.8 | 62.1 |
| Clip-10to1 | 82.5 | 65.5 | 45.3 | 81.5 | 66.3 | 85.8 | 72.1 | 55.2 | 69.3 |
| Clip-14to1 | 81.5 | 68.2 | 43.2 | 81.8 | 68.1 | 86.6 | 70.1 | 51.4 | 68.9 |
| Window-5 | 69.0 | 63.2 | 38.7 | 72.8 | 67.7 | 85.8 | 69.0 | 47.7 | 64.2 |
| Window-6 | 74.0 | 68.2 | 40.0 | 71.3 | 70.8 | 85.3 | 66.9 | 45.8 | 65.3 |
| Window-7 | 70.0 | 64.5 | 40.6 | 71.3 | 65.0 | 86.0 | 70.2 | 42.8 | 63.8 |
| Sum-12to1 | 78.0 | 71.0 | 46.5 | 80.5 | 64.6 | 88.9 | 70.6 | 50.4 | 68.8 |

## N. Environment Specifications

Different environments have varying observation lengths per turn, which affects the total context consumed. To ensure fair cross-environment comparison, we set environment-specific maximum steps that normalize for these differences, resulting in comparable context budgets across tasks. The relative step limits across environments follow the established setup from AgentGym (Xi et al., 2024), scaled proportionally (4×) from expert trajectory lengths to provide sufficient exploration headroom. For full history baselines, we set maximum history windows adapted to each environment's observation length to enable maximum context utilization while respecting practical context limits.

**WebShop(WS)**: WebShop (Yao et al., 2022) is a simulated e-commerce platform where agents execute product procurement tasks adhering to predefined criteria through interface interactions. The environment integrates 12,000 structured instructions and leverages over one million real-world Amazon product listings, with 6,910 instructions selected for task execution. Agents can navigate through button-based interactions or utilize text-based search functionality to locate and purchase products meeting specific requirements. Performance is quantified via average score, with task sequences limited to 40 rounds to balance efficiency and practical applicability. When configured with full history strategy, we set the maximum history window to 35. We utilize the AgentGym library while maintaining the original system prompt, and format available actions as attachments to environment observations to provide comprehensive action space information.

For WebShop evaluation using GPT-4o-mini, we found that some product descriptions containing sensitive content caused

evaluation model refusals. To ensure fair comparison, we controlled both methods to use nearly identical feasible evaluation samples.

**Maze(MZ)**: Maze is a grid-based navigation task from LMRL Gym (Abdulhai et al., 2025) where agents must reach fixed goal locations through strategic movement. At each step, agents move one cell in four directions (up, down, left, right), with observations indicating current position, goal position, and adjacent wall information. The environment supports both fully observable and partially observable variants, with the latter providing only action history. Evaluation employs binary success metrics: episodes score 1 if agents reach goals within step budgets and 0 otherwise. Maximum episode length is set to 60 steps, with identical limits for full history configurations. The deterministic nature of movement mechanics ensures consistent evaluation across different context management strategies while testing spatial reasoning and pathfinding capabilities.

**ALFWorld(ALF)**: ALFWorld (Shridhar et al., 2021) extends the TextWorld framework to household settings, requiring agents to navigate rooms and perform everyday activities involving common-sense reasoning. Tasks encompass diverse textual actions including object manipulation (picking up, placing items), furniture interaction, and environmental inspection. Each action undergoes validation against rule-based simulators providing textual feedback reflecting updated world states. Agent performance is evaluated using success rates, with episodes capped at 120 rounds and maximum full history length of 50. We employ the AgentGym library preserving the original system prompt to maintain consistency with established benchmarks while ensuring fair comparison across different context management approaches.

**TextCraft(TC)**: TextCraft is a text-only environment designed around Minecraft crafting mechanics, providing controlled settings for evaluating compositional reasoning and planning capabilities. The environment constructs crafting trees consisting of 544 nodes, each corresponding to craftable target items. For each task, environments specify target items alongside crafting command sequences derived from trees. Agents issue three action types: `craft <item> using <ingredients>`, `get <item>`, and `inventory`. Agents receive rewards of 1 only upon successfully crafting specified target items. Performance measurement uses success rates with maximum episode lengths capped at 80 steps. For full history experiments, maximum history windows are set to 60. We utilize the AgentGym library while adopting system prompts from (Prasad et al., 2024) with minor modifications to ensure compatibility.

**2048**: Game2048 appears in the TextArena suite (Guertler et al., 2025) as single-player logic puzzles adapted to text-based frameworks, originally inspired by sliding tile games. Tasks evaluate agents' abilities to reason over board states, tile merging mechanics, and long-term planning strategies. Environments simulate 4×4 grids where cells hold tile values as powers of two. Agents make moves in four directions causing tiles with identical values to merge into double-value tiles, thereby increasing game scores. We adapt evaluations for AgentGym framework compatibility while preserving TextArena's official scoring criteria with task sequences limited to 60 rounds and maximum history windows of 20 for full history strategies. Reward systems provide +1.0 for successfully reaching target tiles (default 2048). When games end without reaching targets, partial rewards are computed using weighted formulas: 50% based on score progress and 50

**FrozenLake(FL)**: FrozenLake adapts standard Frozen Lake grids from OpenAI Gym into TextArena's text-based framework. Agents navigate grids from start positions to goals ("G") while avoiding holes ("H") and traversing frozen tiles (empty spaces). Agent starting positions are marked as "P" and can begin from any corner. Actions correspond to four discrete directions with observations provided as visual text-based grid representations showing current board states with player positions. Unlike OpenAI Gym versions, our implementation excludes slippery surface mechanics—all movements are deterministic and execute exactly as commanded. Agents receive +1.0 rewards upon reaching goals. When falling into holes or exceeding step limits, partial rewards are calculated based on progress toward goals using BFS-computed shortest path distances. Episodes terminate upon reaching goals, falling into holes, or exceeding 40-round step limits. For full history experiments, maximum history windows are set to 20.

**Hangman(HM)**: Hangman implements classic word-guessing games within TextArena's text-based framework. Agents must deduce hidden English words by guessing individual letters or attempting complete words. Words are initially displayed as underscores ("_") with each underscore representing one letter. Game boards show column numbers (C00, C01, etc.) above letter positions for reference. Actions consist of single letter guesses formatted as "[L]" or complete word guesses as "[WORD]" (case-insensitive). When correct letters are guessed, all instances are revealed in respective positions. Agents maintain visible histories of previously guessed letters to avoid repetition. Observations include current board states showing revealed letters and underscores, remaining try counts, and already guessed letter sets. Reward systems provide +1.0 for successfully guessing complete words and partial rewards (0.0-1.0) calculated as ratios of correctly revealed letter positions to total word lengths when agents reach step limits. We adapt evaluations for AgentGym framework compatibility while

preserving TextArena's official scoring criteria with task sequences limited to 40 rounds.

**RushHour(RH)**: Rush Hour is a classic sliding block puzzle game adapted as text-based environments for evaluating spatial reasoning and sequential planning abilities. Environments present 6×6 grid boards with vehicles of varying lengths (cars of length 2, trucks of length 3) positioned horizontally or vertically. Objectives involve maneuvering designated red cars (marked as 'X') to exits located at right board edges by sliding other vehicles out of paths. In each episode, agents interact with randomly generated puzzle configurations of varying difficulties (easy, medium, hard) determined by backward scrambling move numbers applied from solved states. Action spaces consist of vehicle movement commands formatted as [`vehicle_id``direction`], where `vehicle_id` represents capital letters (A-Z, with X representing red cars) and directions are '+' (forward/right for horizontal vehicles, down for vertical vehicles) or '-' (backward/left for horizontal vehicles, up for vertical vehicles). Vehicles move only along orientation axes and cannot overlap. Agents receive 1.0 rewards upon successfully guiding red cars to exit positions. For episodes reaching maximum step limits without solving puzzles, partial rewards are determined by red car proximity to exits. Performance is measured by success rates across puzzle difficulties with maximum episode lengths set to 50 steps and maximum history windows of 50 for full history strategies.

# O. Prompt Construction

We structure multi-turn conversations using a sequential message format where each interaction cycle consists of system instructions, user goals, observations, and assistant actions. The conversation flow follows this pattern:

1. `system`: Environment-specific instructions and task descriptions

2. `user`: Initial goal or task specification

3. `user`: Current observation from environment

4. `assistant`: Generated thought and action based on observation

5. `user`: Next observation after action execution

6. `assistant`: Subsequent thought and action, and so on...

For observation formatting, we implement a two-part structure instead of direct text inclusion. Each observation is split into: (1) a structured header `user("Observation:\n")` followed by (2) the actual observation content `user(obs_content)`. This explicit formatting significantly improves Qwen3-8B's comprehension and task performance by providing clear semantic boundaries between different types of information.

# P. Comparison with StreamingLLM and Retrieval-Based Baselines

## P.1. StreamingLLM Baseline

We compare Clip Context against StreamingLLM (Xiao et al., 2024), which maintains efficiency by retaining a small set of attention-sink tokens and a fixed sliding window of recent tokens. We implement StreamingLLM with `num_sink_tokens=4` and `context_length=2048` under the same long-context agent setting (Qwen3-8B). Results are shown in Table 11.

*Table 11.* StreamingLLM vs. Clip Context (Qwen3-8B). Success rates (%) are reported.

| Method | MZ | ALF | WS | TC | FL | HM | 2048 | RH | Avg |
|---|---|---|---|---|---|---|---|---|---|
| Long + StreamingLLM | 34.0 | 56.5 | 26.7 | 63.5 | 67.9 | 77.5 | 66.4 | 44.1 | 54.6 |
| Window-6 | 74.0 | 68.2 | 40.0 | 71.3 | 67.5 | 85.3 | 66.9 | 45.8 | 64.9 |
| Clip-12to1 | 83.0 | 67.7 | 44.4 | 82.3 | 67.5 | 85.1 | 70.9 | 50.2 | 68.9 |

Both Window Context and Clip Context outperform StreamingLLM. We attribute this to two factors. First, Clip is designed to mitigate the inertia problem specific to agent scenarios, while StreamingLLM is designed for inference efficiency. Clip periodically clears context to balance exploration and exploitation, whereas StreamingLLM maintains a fixed-size sliding window that only indirectly reduces diagonal attention strength. Second, round-level truncation is more suitable for agent scenarios than token-level truncation. Both Window and Clip drop complete interaction turns, preserving round boundaries.

StreamingLLM discards tokens at the token level, which breaks agent-round boundaries and results in incomplete turns whose number varies across environments due to differing observation lengths.

### P.2. Retrieval-Based Baseline (SeCoM)

We compare Clip Context against SeCoM (Pan et al., 2025), a retrieval-based memory method that retrieves the two most similar historical observation–action pairs based on the current observation and appends them to the context. To ensure a fair comparison, we control both methods to receive the same average input length (Table 12).

*Table 12.* Clip Context vs. SeCoM retrieval-based method (Qwen3-8B). Success rates (%) are reported. SeCoM is paired with both Window and Clip context strategies.

| Method | MZ | ALF | WS | TC | FL | HM | 2048 | RH | Avg |
|---|---|---|---|---|---|---|---|---|---|
| SeCoM + Window | 44.0 | 65.5 | 39.1 | 75.0 | 62.5 | 83.9 | 72.5 | 46.5 | 61.1 |
| SeCoM + Clip | 55.5 | 60.0 | 42.8 | 76.0 | 66.1 | 84.5 | 68.3 | 47.1 | 62.5 |
| Clip-12to1 | 73.0 | 69.0 | 45.6 | 82.5 | 66.2 | 84.2 | 73.3 | 43.0 | 67.1 |

Clip Context outperforms both SeCoM variants. Directly retaining the latest rounds yields better results than SeCoM retrieval, mainly because recent information is unaffected by retrieval query quality and does not depend on environment-specific question formulation or retrieval design. This result is consistent with findings for summarization in Table 1, confirming that Clip Context is a simple and robust baseline for context management.

Clip Context and retrieval-based methods address different aspects of context management and are complementary. Clip focuses on reducing inertia through context clearing, while retrieval methods focus on supplementing context with relevant past experience. Combining the two is a promising direction for tasks with strong long-range dependencies.

## Q. Base Model Inertia Comparison and CPL Generalization

### Q.1. Base Model Inertia Comparison

The severity of conversational inertia varies across models. Table 13 compares inertia indicators for Llama3.1-8B-Instruct and Qwen3-8B. Qwen3-8B exhibits higher diagonal attention, higher assistant token attention, and substantially higher repeat-last-action rates, indicating stronger inertia in its base behavior. This explains why CPL and Clip yield larger gains on Qwen3-8B than on Llama3.1-8B: models with stronger inertia benefit more from inertia-targeted methods.

*Table 13.* Base model inertia indicators. Diagonal attention and assistant attention are averaged across 8 environments.

| Model | Diag. Attn | Asst. Attn | Repeat Last (MZ) | Repeat Last (TC) |
|---|---|---|---|---|
| Llama3.1-8B-Instruct | 0.0154 | 0.0713 | 17.02% | 6.02% |
| Qwen3-8B | 0.0205 | 0.0901 | 22.16% | 28.05% |

### Q.2. CPL Generalization to Qwen3-1.7B

To evaluate whether CPL generalizes beyond the training model family, we train Qwen3-1.7B using preference data collected from the Qwen3-8B family. Table 14 shows that CPL improves Qwen3-1.7B performance across environments, demonstrating that the preference signal generalizes to smaller models.

*Table 14.* CPL applied to Qwen3-1.7B using preference data from Qwen3-8B. Evaluation under Window-6 context strategy. Success rates (%) are reported.

| Model | MZ | ALF | WS | TC | FL | HM | 2048 | RH | Avg |
|---|---|---|---|---|---|---|---|---|---|
| Qwen3-1.7B | 47.5 | 36.2 | 50.3 | 38.5 | 43.9 | 83.2 | 67.4 | 40.2 | 50.9 |
| Qwen3-1.7B CPL | 54.0 | 38.0 | 48.0 | 43.0 | 44.4 | 84.7 | 70.3 | 38.8 | 52.7 |

# R. Attention Heatmap Visualizations Across Environments

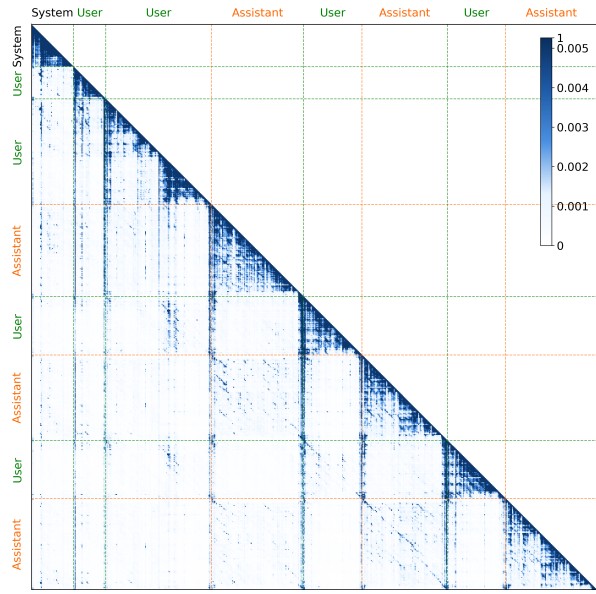

*Figure 13.* 2048 Environment Attention Patterns

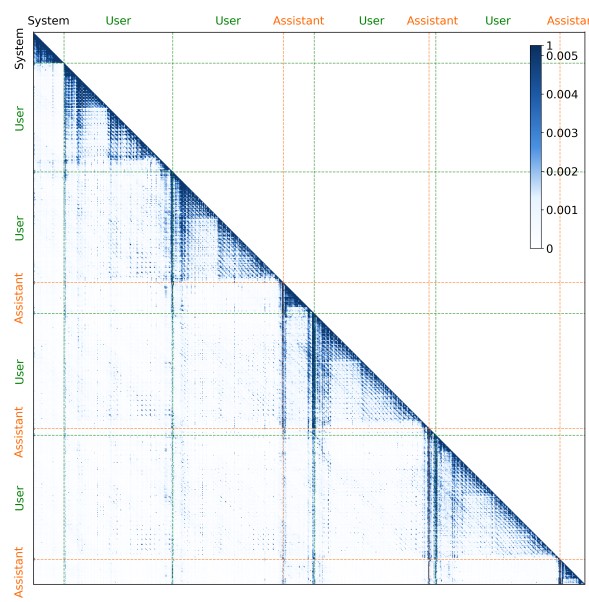

*Figure 14.* ALF Environment Attention Patterns

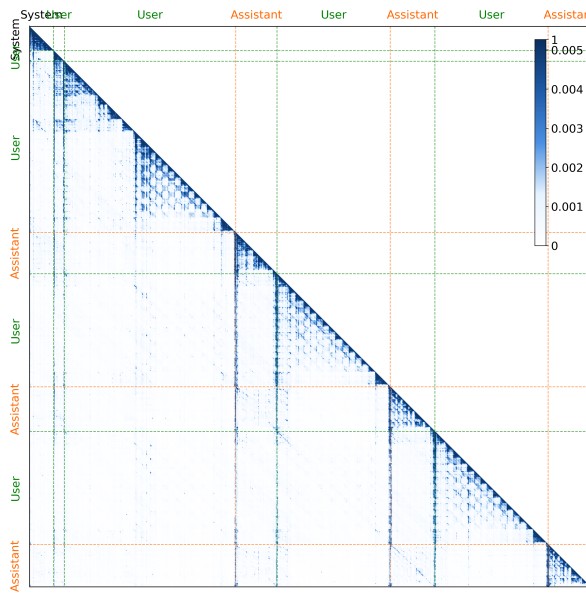

*Figure 15.* FrozenLake Environment Attention Patterns

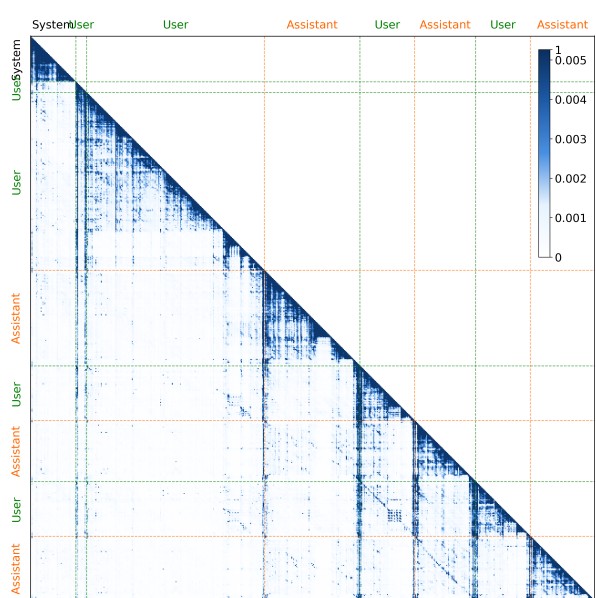

*Figure 16.* Hangman Environment Attention Patterns

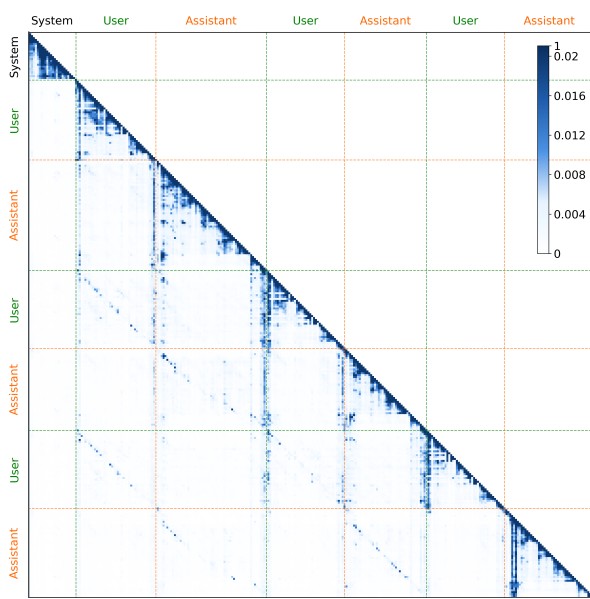

*Figure 17.* Maze Environment Attention Patterns

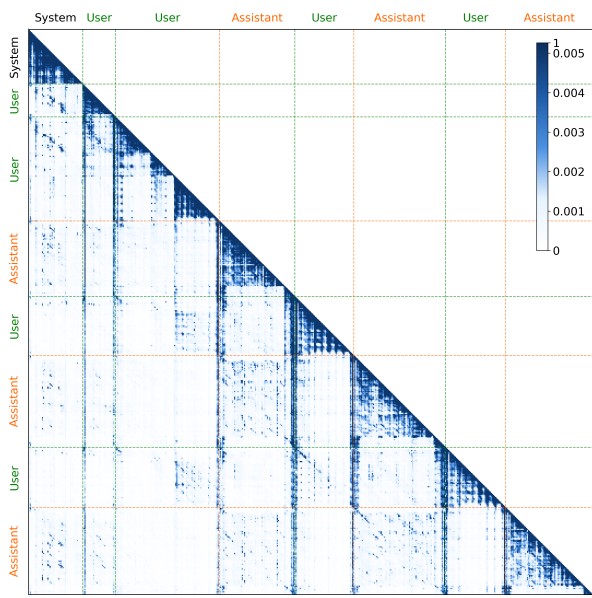

*Figure 18.* RushHour Environment Attention Patterns

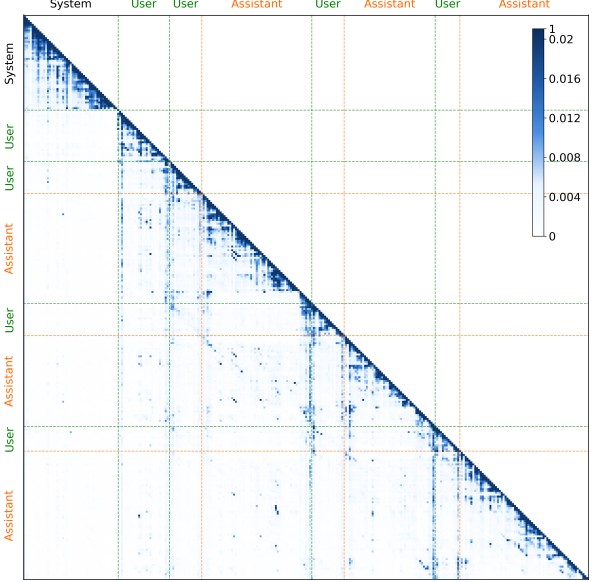

*Figure 19.* TextCraft Environment Attention Patterns

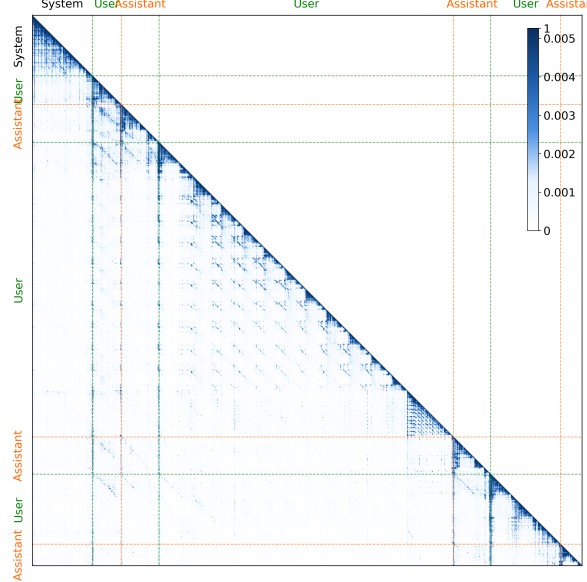

*Figure 20.* WebShop Environment Attention Patterns

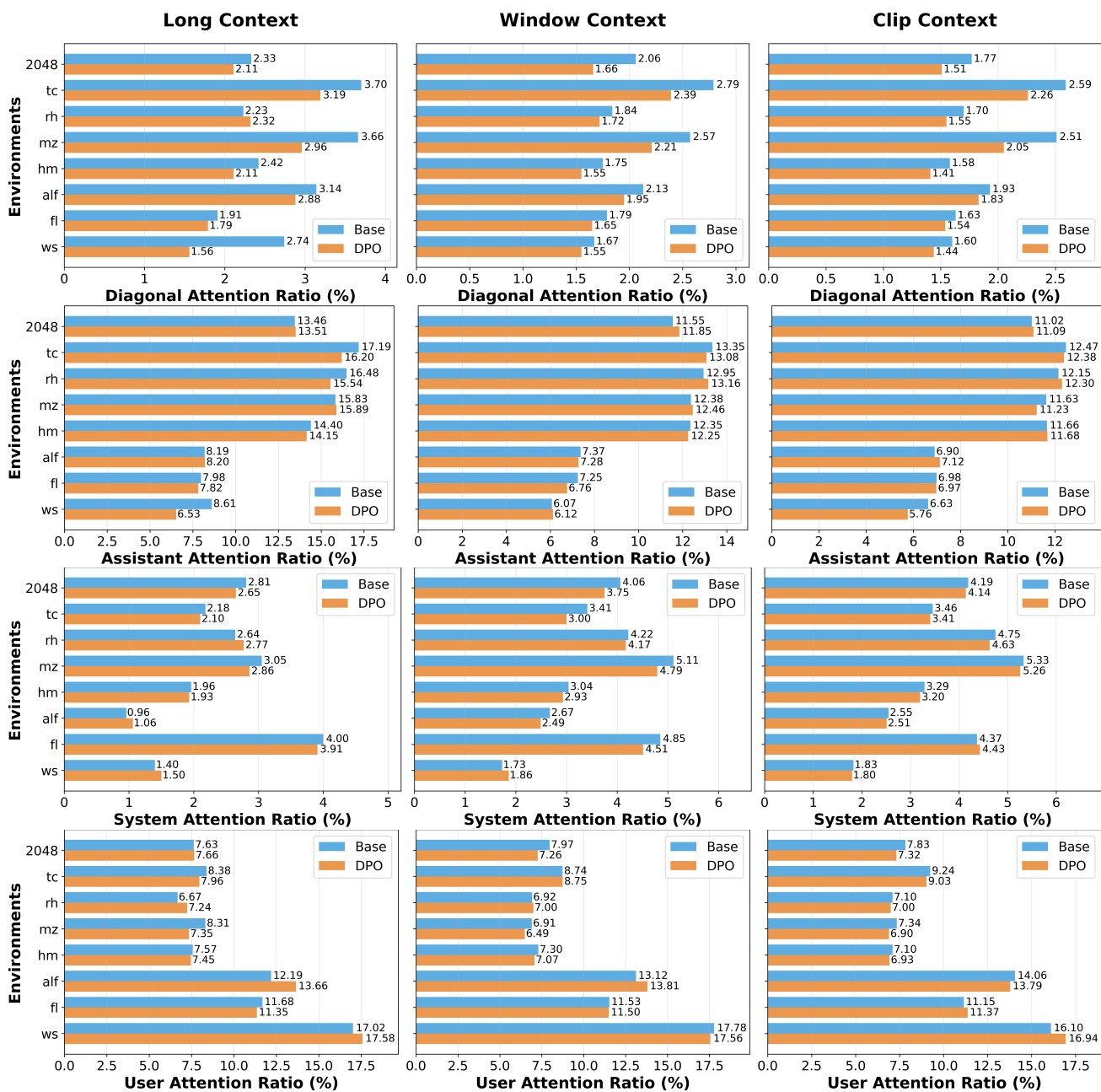

*Figure 11.* Attention distribution analysis across different component types and context management strategies. From top to bottom: diagonal attention (self-attention to previous assistant responses), assistant attention (attention to all assistant tokens), system attention (attention to system prompt tokens), and user attention (attention to user input tokens). Each subplot compares base models and CPL-trained models across three context management approaches: Long (full context), Window (recent context only), and Clip (filtered context).

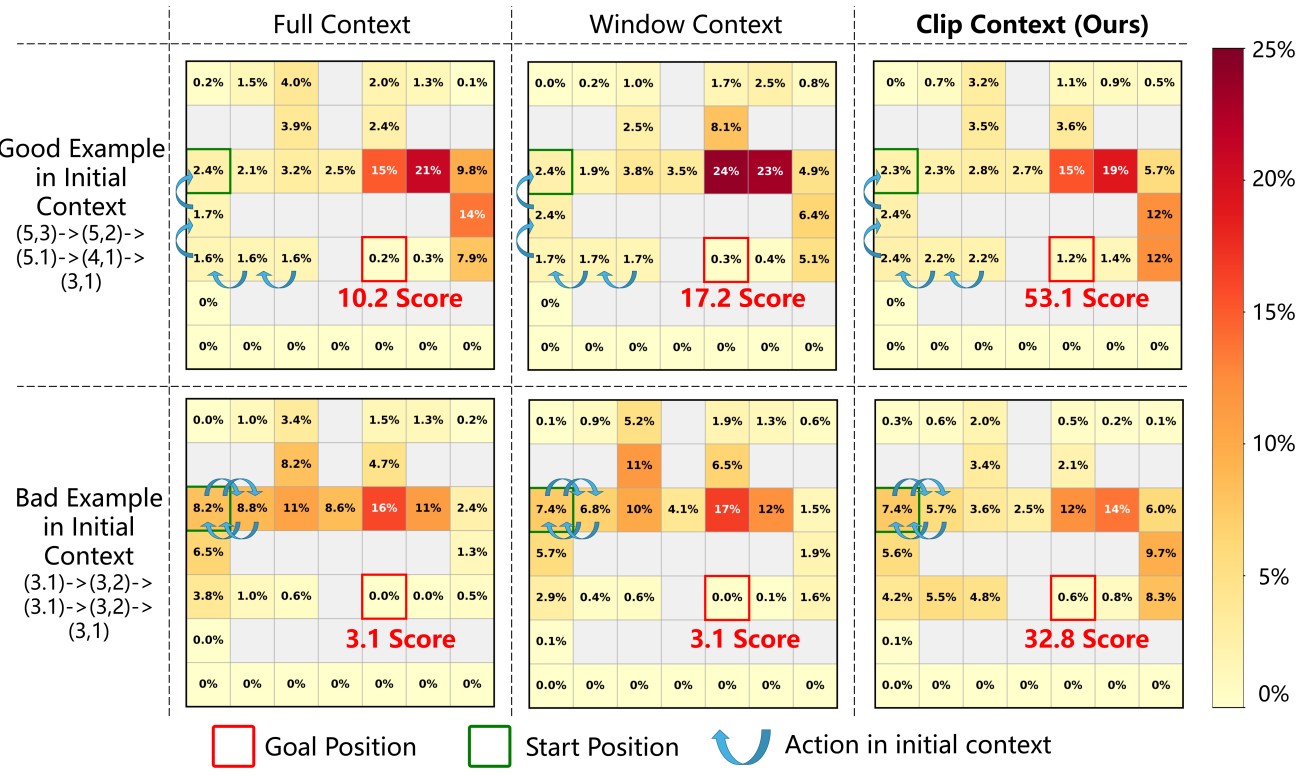

*Figure 12.* Context influence analysis in maze environment. We compare three context management methods (Full context, Window context, and Clip context) under both good and bad initial contexts. Each heatmap shows visit frequency as percentages (e.g., 0.3% means the agent spent 0.3% of total steps at that position). Scores in the tables indicate overall task performance.

