# OpenReview forum: "Mitigating Conversational Inertia in Multi-Turn Agents"
_ICML.cc/2026/Conference — ICML 2026 regular_

### Official Review · Reviewer_XMmw · 2026-02-15

**Soundness:** 2
**Presentation:** 3
**Significance:** 3
**Originality:** 3
**Overall Recommendation:** 4
**Confidence:** 4

**Summary:**

The paper propose a DPO-like method: Context Preference Learning (CPL) to address conversational inertia, a phenomenon where the model exhibit strong diagnoal attention to previous responses, which is associated with imitation bias that constrains exploration. It collects preference pair based on one key insight that actions generated with longer context exhibit stronger inertia than those with shorter context. Furthermore, the paper introduces a simple method by retaining L historical turns every H step to clip the context. The experimental results show limited improvement and improved efficiency.

**Compliance With Llm Reviewing Policy:**

Affirmed.

**Final Justification:**

The rebuttal addressed my concerns and provided additional results.

**Key Questions For Authors:**

See above

**Limitations:**

Yes

**Strengths And Weaknesses:**

Strengths:

1. The targeted problem conversational inertia is important and valueable.

Weaknesses

1. The key insight to collect such preference data is not well supported. It is not reasonable to use results to motivate the start point. Also, it is better to indicate when does it hold and not?
2. The performance gain is limited and marginal. It is important to directly compare base model and CPL-trained model, and as shown in Table 1, the performance gains on Lllama3.1-8B is limited, less than 1 point and even 0.5 point, even without considering randomness, so does Figure 6.
3. The proposed clip method is incremental, there are tons of summarization-based methods or simply keeping latest L step every H step. It also does not directly connnect with the target problem.

---

> ### Author Rebuttal · Authors · 2026-03-31
>
> ## Response to Reviewer XMmw
>
> ### W1: How is the key insight for preference data collection supported?
>
> The reviewer raises two concerns: (a) using results to motivate the starting point, and (b) when does the assumption hold.
>
> **(a) On motivation.** We did not construct the preference data based on task performance. Instead, we first discovered the high diagonal attention phenomenon through attention analysis (Figure 2), then used it as the key insight for data collection. The preference pairs are constructed by pairing actions generated under long context (high inertia) against those under moderate context (lower inertia). The preferred actions in DPO data come from Window context. After CPL training, the model outperforms Window context. This demonstrates that CPL does not simply fit the preferred Window outcomes, but improves performance by optimizing the inertia gap through preference pairs.
>
> **(b) When does it hold.** Our assumption is that actions from excessively long contexts exhibit stronger inertia than those from moderate contexts. This assumption holds broadly across our evaluated environments. Figure 2 shows that diagonal attention increases consistently as context rounds grow in Maze environment. Table 1 confirms that moderate context (Window) outperforms excessively long context (Long) in nearly all environments.
>
> ### W2: Why is the CPL performance gain on Llama3.1-8B limited?
>
> Inertia is a general phenomenon across models, but its severity varies. As shown in R-Table 4, Llama3.1-8B exhibits lower diagonal attention, lower assistant attention, and lower repeat-last-action rates than Qwen3-8B, indicating that Llama3.1-8B already has relatively mild inertia in its base behavior. CPL and Clip reduce inertia, so their gains naturally scale with the severity of the problem. This is consistent with the expectation that models with stronger inertia benefit more from inertia-targeted methods. Notably, Clip also improves GPT-4o-mini by +5.1 over Window (Table 1) by reducing inertia, confirming that this mechanism generalizes beyond open-source models.
>
> **R-Table 4: Base model inertia comparison.** Diagonal attention and assistant attention are averaged across 8 environments.
>
> | Model | Diag Attn | Assistant Attn | Repeat Last Action (Maze) | Repeat Last Action (Textcraft) |
> |---|---|---|---|---|
> | Llama3.1-8B | 0.0154 | 0.0713 | 17.02% | 6.02% |
> | Qwen3-8B | 0.0205 | 0.0901 | 22.16% | 28.05% |
>
> We trained Qwen3-1.7B with CPL using preference data that favors lower-inertia actions collected from the Qwen3-8B family, and observed clear improvements as shown in R-Table 5.
>
> **R-Table 5: CPL on Qwen3-1.7B**
>
> | Model | MZ | ALF | WS | TC | FL | HM | 2048 | RH | Avg |
> |---|---|---|---|---|---|---|---|---|---|
> | Qwen3-1.7B | 47.5 | 36.2 | 50.3 | 38.5 | 43.9 | 83.2 | 67.4 | 40.2 | 50.9 |
> | Qwen3-1.7B CPL | 54.0 | 38.0 | 48.0 | 43.0 | 44.4 | 84.7 | 70.3 | 38.8 | 52.7 |
>
> ### W3: Is the Clip method incremental? How does it relate to inertia?
>
> The Clip method is not incremental. We highlight two aspects:
>
> 1. **Clip addresses a different problem than summarization.** Clip is a context-clearing strategy designed to reduce inertia, while summarization methods are compression-and-restructuring strategies. These address different factors of context management, and we do not claim that Clip solves all context management problems.
>
> 2. **Clip isolates the context-clearing effect from summarization methods.** To the best of our knowledge, existing summary-based methods (ReSum) simultaneously perform summarization and context clipping, coupling context-clearing effects with compression-and-restructuring effects. They do not reveal the importance of context-clearing alone. Our framework systematically compares Window, Clip context-clearing strategies and shows that Clip alone achieves comparable or better performance than ReSum across 8 environments (Table 1). This also helps explain why summarization sometimes fails to outperform simpler strategies [3].
>
> **Clip directly connects to the target problem of mitigating inertia.** Clip discards historical context, which results in information loss but reduces inertia. Our experiments consistently show that the performance gain from reduced inertia outweighs the loss from discarded information (Table 1: Clip outperforms Long and Window). This confirms the importance of addressing inertia in multi-turn agents.
>
> > [3] Deng et al., "The Complexity Trap: Simple Observation Masking Is as Efficient as LLM Summarization for Agent Context Management", 2025.
>
> ### On efficiency
>
> Theoretically, Clip reduces prefill-stage computation by a factor of $W$ (Appendix F). In practice, the overall latency includes both prefill and decoding stages. In our experiments, Clip reduces prefill-stage latency by 70%. The decoding stage is primarily determined by hardware bandwidth and output token length, which is not the focus of our claim.

---

> > ### Author Rebuttal · Reviewer_XMmw · 2026-04-02
> >
> > Thanks authors for the reply. I have increased my score accordingly. :)

---

### Official Review · Reviewer_MgG5 · 2026-03-02

**Soundness:** 3
**Presentation:** 3
**Significance:** 3
**Originality:** 3
**Overall Recommendation:** 4
**Confidence:** 2

**Summary:**

The paper tackles conversational inertia problem that exists today in LLMs. Through attention visualization the author remark that models exhibit strong diagonal attention to previous assistant outputs, revealing phenomenon associated with conversational inertia. To answer this problem, they introduce CPL, Context Preference Learning, a training method that forces the model to generate two actions $a^{long}$, using full context history, and $a^{short}$, using only recent context. While they execute $a^{long}$ in the environment to get the next action, they preference datasets only uses the shorter context as training input. Finally they formulate three context control strategies: Full, Window, Clip. Though evaluation on benchmarks they show that the combination of CPL + Clip performs better.

**Compliance With Llm Reviewing Policy:**

Affirmed.

**Final Justification:**

The rebuttal reinforced my prior assessment on soundness and significance, while addressing my concerns on originality which changed my evaluation on this component. However, I still feel that this paper deserves a weak accept as it perfectly matches the description "Technically solid paper that advances at least one sub-area of AI, with a contribution that others are likely to build on.", but doesn't provide more.

**Key Questions For Authors:**

Could you please summarize you own assessment of the originality (1), and limitations (2) of your methods.

**Limitations:**

No, the related work and discussion sections could have been more precise in exposing expose the significance, originality and limitations of the method.

**Strengths And Weaknesses:**

**Soundness :**

[+] The proposed method is sound, the claims are supported by experimental results. The results provided rely on a priori known Benchmarks, but I don't have the necessary expertise to confirm this.

[-] I don't see any claim that can be properly derived from Subsection 3.9 on hyperparameter analysis as taking the average on 8 environment doesn't really makes much sense. It seems to claim that having a small $L$ and high $H$ is better which, in some sense, contradicts Table 2.

**Presentation :**

[+] The submission is globally clearly written and well structured. As acknowledged by the author, having LLM to review the writing and correct the wording of subsections is relevant.

[-] I would have preferred to have the related work section after the introduction as motivations for the paper. Section 10, impact statement isn't really relevant as they isn't any direct ethical considerations or societal implications different from the whole "LLM field".

**Significance :**

[$\pm$] The paper address an important problem. The method proposed seems to be an advanced in machine learning, however more precise related work/discussion/limitations sections would have helped assess the significance of the method.

**Originality :**

[$\pm$] Same remark as significance, the two methods CPL and clip seem original but more precise related work/discussion/limitations sections would have helped assess the originality of the method.

---

> ### Author Rebuttal · Authors · 2026-03-31
>
> ## Response to Reviewer MgG5
>
> ### W1: per-environment hyperparameter analysis and its relationship with Table 2
>
> To better illustrate our claim we report a per environment result below.
>
> **R-Table 3: Clip vs Window on eight per-environment results (Qwen3-8B)**
>
> | Method | MZ | ALF | WS | TC | FL | HM | 2048 | RH | Avg |
> |---|---|---|---|---|---|---|---|---|---|
> | Clip 12to1 | 83.0 | 67.7 | 44.4 | 82.3 | 69.7 | 85.1 | 70.9 | 50.2 | 69.2 |
> | Clip 12to3 | 73.0 | 69.0 | 45.6 | 82.5 | 66.2 | 84.2 | 73.3 | 43.0 | 67.1 |
> | Clip 12to6 | 53.0 | 67.5 | 42.4 | 76.5 | 65.0 | 80.5 | 67.4 | 44.8 | 62.1 |
> | Clip 10to1 | 82.5 | 65.5 | 45.3 | 81.5 | 66.3 | 85.8 | 72.1 | 55.2 | 69.3 |
> | Clip 14to1 | 81.5 | 68.2 | 43.2 | 81.8 | 68.1 | 86.6 | 70.1 | 51.4 | 68.9 |
> | Window 5 | 69.0 | 63.2 | 38.7 | 72.8 | 67.7 | 85.8 | 69.0 | 47.7 | 64.2 |
> | Window 6 | 74.0 | 68.2 | 40.0 | 71.3 | 70.8 | 85.3 | 66.9 | 45.8 | 65.3 |
> | Window 7 | 70.0 | 64.5 | 40.6 | 71.3 | 65.0 | 86.0 | 70.2 | 42.8 | 63.8 |
> | Sum 12to1 | 78.0 | 71.0 | 46.5 | 80.5 | 64.6 | 88.9 | 70.6 | 50.4 | 68.8 |
>
>
> Our claims are as follows:
>
> 1. Clip context is more robust than Window context. At the same average input length, as R-Table 3 shows, varying the Clip hyperparameters (Clip 10to1, 12to1, 14to1) leads to nearly identical average scores across 8 environments, while Window variants (Window 5, 6, 7) fluctuate more. This makes Clip a safer default for new applications. In Subsection 3.9 and Table 1, we use the average specifically to demonstrate that Clip does not require per-environment hyperparameter tuning to outperform Window methods.
> 2. A higher L parameter is more suitable for tasks with long-range dependencies.
> 3. The upper limit H provides more contextual information for decision-making, and thus should not be set too small.
>
> **No contradiction with Table 2.** The optimal hyperparameters differ across environments, as R-Table 3 shows. Table 2 reports results on BrowseComp, while Subsection 3.9 shows an average over 8 simulator-based environments. Similar to ALFWorld and WebShop, BrowseComp has strong long-range dependencies where the agent must synthesize information gathered across many browsing steps. Therefore, Clip 12to6 performs best in BrowseComp, while Clip 12to1 is more suitable on average across these eight simulated environments. We also provide principled hyperparameter selection guidelines in our response to Reviewer ZCfo Q1.
>
> ### Q2: Limitations and Related Work
>
> We appreciate the suggestion to more precisely contextualize our contributions.
>
> **Related work positioning.** Prior context management methods fall into three categories: sliding window, summarization-based, and retrieval-based. Sliding window methods (e.g., StreamingLLM) maintain a fixed-size context but do not address inertia accumulated within the window. Summarization methods (e.g., ReSum, AgentFold) replace raw history with compressed summaries. Retrieval methods (e.g., SeCoM, MemR3) add relevant past experiences to context, but require environment-specific retrieval design and are less generalizable. All three categories treat performance degradation as a consequence of context length or information overload, without isolating the role of self-imitation bias in the model's own outputs. Our work identifies conversational inertia as a distinct failure mode (Figure 2, R-Table 4) and proposes both a training method (CPL) and an inference strategy (Clip) that target this mechanism.
>
> Summary-based methods focus on compression and information restructuring, while Clip focuses on context clearing to reduce inertia. These two directions address different factors of context management [1, 2]. Recent work also recognizes selective forgetting as an independent design axis (MemoryAgentBench, FadeMem). Clip alone achieves comparable performance to ReSum across 8 environments (R-Table 3, Subsection 3.9, Appendix M), confirming that inertia reduction is an independently effective mechanism. Our methods are orthogonal to summarization and retrieval and can be combined with them (R-Table 2).
>
> **Limitations.** We acknowledge the following limitations and will add an explicit paragraph to the main text. (1) Clip inherently involves an information-inertia tradeoff and can cause failures in tasks with strong long-range dependencies and short observations (see Appendix L for failure case analysis). (2) The optimal Clip parameters vary across environments, though moderate defaults (L=1--3, H=2xW) work broadly without per-environment tuning. (3) Our analysis focuses on conversational inertia as one contributor to multi-turn degradation. Other factors such as error propagation and reward sparsity are not addressed in this work.
>
> > [1] Deng et al., "The Complexity Trap: Simple Observation Masking Is as Efficient as LLM Summarization for Agent Context Management", 2025.
> > [2] Levy et al., "Context Length Alone Hurts LLM Performance Despite Perfect Retrieval", 2025.

---

> > ### Author Rebuttal · Reviewer_MgG5 · 2026-04-02
> >
> > I thank the authors for their reply. I have adjusted the originality score accordingly.

---

### Official Review · Reviewer_ZCfo · 2026-03-12

**Soundness:** 3
**Presentation:** 3
**Significance:** 3
**Originality:** 2
**Overall Recommendation:** 5
**Confidence:** 3

**Summary:**

This paper studies conversational inertia in multi-turn LLM agents: as context grows, models tend to over-attend to and imitate their own earlier responses, which can hurt performance. To mitigate this, the authors propose a Context Bias Calibration framework with two parts: Context Preference Learning, which uses preference-based fine-tuning to favor actions produced from less biased context, and clip context, an inference-time strategy that periodically clears older history to reduce inertia.

The paper supports this with attention analysis and experiments across a range of agentic environments, including a deep-research setting. The main contribution is identifying conversational inertia as a distinct failure mode and introducing both a training-time and an inference-time method to reduce it.

**Compliance With Llm Reviewing Policy:**

Affirmed.

**Final Justification:**

My main concerns are addressed, so I raise my score to accept.

**Key Questions For Authors:**

1. The paper provides a hyperparameter ablation for the clipping schedule, but it is still unclear how one should choose the parameters for a new environment in practice. Is there any general rule for selecting a good clipping range based on observable task properties, or does this ultimately require trying several parameter settings and comparing performance empirically?
2. I wonder whether the relatively weak performance of summarization based methods may partly depend on the environments used in the paper. Many of the evaluated environments seem close to Markovian in the sense that the current state already contains most of the information needed for the next decision, so clipping history may not incur much information loss. In contrast, in environments such as ALFWorld, which involve more long term planning and delayed dependencies, summarization based methods appear to perform better than clipping. Could the authors comment more explicitly on what kinds of environments favor clipping versus summarization?
3. The paper compares against long, window, and summarization based context management, but it would be useful to understand how clip context compares with stronger explicit memory or retrieval based approaches. Do the authors believe the proposed method is complementary to such methods, or mainly an alternative to them?

**Limitations:**

The paper does not include a dedicated paragraph in the main text that discusses its limitations in detail. That said, the authors do acknowledge several limitations of the current work, including the disclaimer in Appendix G. It would improve the paper if the authors could add a more explicit and consolidated discussion of limitations.

**Strengths And Weaknesses:**

**Soundness.** The paper is fairly solid empirically. It does more than report overall score improvements and also tries to explain the failure mode through attention analysis, which makes the story more convincing. The evaluation is also reasonably broad, with multiple models, several agent environments, and a long horizon research setting. That said, the evidence on mechanism is still not fully decisive. The paper shows that certain attention patterns go together with worse behavior and become weaker under the proposed methods, but this still falls short of proving that they are the main cause of the problem. Also, clip context helps partly by removing history, so there is an obvious tradeoff between reducing inertia and keeping useful information. The paper discusses this, but does not fully pin down when the method should help and when it may hurt.

**Presentation.** The paper is generally clear and easy to read. The setup, the two proposed components, and the comparison among long, window, and clip settings are all presented in a sensible order.

**Significance.** The paper studies a real problem for multi-turn agents. In practice, having more interaction history does not always help, and this is an important point. The proposed clip context method is simple enough that people could actually use it in existing systems, and the efficiency gain makes the contribution more useful in practice. At the same time, the method should be seen more as a practical fix than as a full answer to long horizon memory. It is likely to be most useful as a context management technique inside agent pipelines, rather than as a general solution for tasks that require reliable retention of specific information across many turns.

**Originality.** The most interesting part of the paper is the framing. Treating conversational inertia as a distinct failure mode in multi turn agents gives the work a clear identity, and the paper builds a fairly coherent solution around that idea. This part feels fresh. In contrast, the method itself is fairly simple. Clip context is a direct intervention, and Context Preference Learning is built on an existing preference optimization framework. So the novelty comes more from the problem formulation and diagnosis than from a deeply new algorithmic idea.

---

> ### Author Rebuttal · Authors · 2026-03-31
>
> ## Response to Reviewer ZCfo
>
> ### Q1: How should one choose Clip hyperparameters for a new environment?
>
> We have provided hyperparameter selection guidelines in Appendix M. In short:
> - **H parameter:** Keep H at the existing Window size W or 2xW.
> - **L parameter:** For tasks with state transitions (e.g., navigation, computer use), set L to a low value (L=1 or L=3). For tasks where each step involves equal reasoning effort (e.g., deep research), set L to a moderate value (L=6).
>
> Clip under most parameter settings outperforms Window, making the method easy to apply. R-Table 3 in our response to Reviewer MgG5 provides per-environment breakdowns that demonstrate Clip shows lower variance across environments. Even without environment-specific tuning, a default setting of (L=1--3, H=2xW) consistently matches or exceeds Window performance, making Clip a safe starting point for new downstream applications.
>
> ### Q2: When does clipping favor over summarization?
>
> Clip is favorable over summarization when the environment has a nearly Markov property (e.g., Maze, Textcraft), where the latest several states contain most of the information needed to decide the next action. When the task has strong long-range dependencies and a weak Markov property (e.g., ALFWorld, BrowseComp), keeping moderate context after clipping (e.g., Clip 12to3) can achieve comparable performance to summarization methods.
>
> For environments that explicitly require very long-range dependencies (e.g., ATM-bench, Li et al., 2025), combining Clip with summarization methods is a promising direction.
>
> ### Q3: Clip vs memory/retrieval-based approaches
>
> Clip is complementary to memory and retrieval methods. Clip focuses on reducing inertia, while memory/retrieval methods focus on compression and reconstruction. Therefore, they can be combined. We implemented SeCoM (Pan et al., 2025), a retrieval-based memory method. Specifically, SeCoM retrieves the two most similar historical observation-action pairs based on the current observation and adds them to the context. We compare Clip with regular Window context under a fair setting where both methods receive the same average input length.
>
> **R-Table 2: Clip vs SeCoM (Qwen3-8B)**
>
> | Method | MZ | ALF | WS | TC | FL | HM | 2048 | RH | Avg |
> |---|---|---|---|---|---|---|---|---|---|
> | SeCoM + Window | 44.0 | 65.5 | 39.1 | 75.0 | 62.5 | 83.9 | 72.5 | 46.5 | 61.1 |
> | SeCoM + Clip | 55.5 | 60.0 | 42.8 | 76.0 | 66.1 | 84.5 | 68.3 | 47.1 | 62.5 |
> | Clip | 73.0 | 69.0 | 45.6 | 82.5 | 66.2 | 84.2 | 73.3 | 43.0 | 67.1 |
>
> Our Clip context method achieves better performance compared to Window Context. Besides, we find that simply keeping the latest three rounds yields better results than SeCoM2. This is mainly because directly retaining recent information is less affected by retrieval query quality, and does not depend on environment-specific question formulation or retrieval design. We observed a similar pattern with summarization in Table 1. Therefore, we find that Clip Context is a simple and robust baseline for context management.
>
> We will add an explicit limitations paragraph to the main text as suggested. Please see the Limitations section in our response to Reviewer MgG5.
>
> ### Comments on Soundness, Significance, and Originality
>
> We thank the reviewer for the thoughtful and balanced assessment.
>
> For when the method helps vs. hurts, we discuss this in Q2 above. Clip favors near-Markov environments, while tasks with long-range dependencies benefit from retaining more context or combining with summarization (R-Table 2).
>
> We agree that the novelty is primarily in problem formulation and diagnosis. We see this as a strength. The attention-based diagnosis led directly to both CPL and Clip, and experiments confirm that both reduce diagonal attention while improving performance.

---

> > ### Author Rebuttal · Reviewer_ZCfo · 2026-04-04
> >
> > Thanks the authors for their reply, and I have raised my score to accept.

---

### Official Review · Reviewer_87nX · 2026-03-13

**Soundness:** 3
**Presentation:** 3
**Significance:** 3
**Originality:** 3
**Overall Recommendation:** 4
**Confidence:** 3

**Summary:**

This study identifies conversational inertia as a challenging drawback of LLMs. To tackle this issue, this paper proposes Context Preference Learning and Clip Context. Through thorough evaluation, the results indicate a significant gain of performance and proactive rate.

**Compliance With Llm Reviewing Policy:**

Affirmed.

**Key Questions For Authors:**

I have no other questions.

**Limitations:**

Yes

**Strengths And Weaknesses:**

Strengths:
(1) The method is simple and elegant. It requires no architectural changes, works with existing LLMs, and is KV-cache compatible, making it easy to deploy in real long-running systems.
(2) Experiments are conducted across multiple datasets with comprehensive evaluation metrics.

Weakness:
(1) To highlight novelty, this paper should include StreamingLLM as baseline systems. (Efficient Streaming Language Models with Attention Sinks)
(2) The context preference learning seems redundant to Clip Context. The author should highlight why it is necessary to present the two methods in the same paper.

---

> ### Author Rebuttal · Authors · 2026-03-31
>
> ## Response to Reviewer 87nX
>
> ### W1: StreamingLLM as a baseline
>
> We implemented StreamingLLM (Xiao et al., 2024) under the long-context agent setting with `num_sink_tokens=4` and `context_length=2048`. Results are shown in R-Table 1.
>
> **R-Table 1: StreamingLLM vs Clip Context (Qwen3-8B)**
>
> | Method | MZ | ALF | WS | TC | FL | HM | 2048 | RH | Avg |
> |---|---|---|---|---|---|---|---|---|---|
> | Long + StreamingLLM | 34.0 | 56.5 | 26.7 | 63.5 | 67.9 | 77.5 | 66.4 | 44.1 | 54.6 |
> | Window 6 | 74.0 | 68.2 | 40.0 | 71.3 | 67.5 | 85.3 | 66.9 | 45.8 | 64.9 |
> | Clip 12to1 | 83.0 | 67.7 | 44.4 | 82.3 | 67.5 | 85.1 | 70.9 | 50.2 | 68.9 |
>
> We find that Clip Context achieves the best performance, and both Clip Context and Window Context outperform StreamingLLM. We attribute this to two factors:
>
> 1. Clip is designed to mitigate the inertia problem in agent scenarios, while StreamingLLM is designed for efficiency. In agent scenarios, we find that the inertia problem is one of the factors that hinder performance. Clip periodically clears context to balance exploration and exploitation, mitigating the inertia problem as steps accumulate. StreamingLLM maintains a fixed number of tokens in its sliding window to accelerate inference, which only indirectly reduces diagonal attention strength. Therefore, Clip achieves better performance in agent scenarios.
>
> 2. Round-level truncation is more suitable than token-level truncation for agent scenarios. Both Window and Clip operate at the round level, dropping complete interaction turns at once. StreamingLLM discards tokens at the token level, which breaks agent round boundaries. Some rounds become incomplete, and the number of retained rounds varies across environments due to differing observation lengths. Therefore, Clip and Window Context are more suitable for agent scenarios. We will discuss StreamingLLM in the revision.
>
> ### W2: CPL seems redundant to Clip Context
>
> CPL is not redundant to Clip Context, as we explain from two aspects.
>
> 1. Both CPL and Clip Context address the same underlying conversational inertia problem identified through our attention analysis, and they work on two complementary aspects. CPL is a lightweight training method that calibrates model preferences to favor low-inertia responses via DPO on long-short context preference pairs. Clip Context is a training-free inference strategy that periodically clears accumulated interaction history. These two methods operate at different stages (training vs. inference), so their effects combine to produce further improvement.
>
> 2. Experimental results in Table 1, Figure 6, and Figure 7 show that each method provides gains. Applying CPL alone improves performance by +2.5 on Window Context (64.9 to 67.4) and reduces diagonal attention by 11.6%. On top of CPL, switching from Window Context to Clip Context further improves performance by +5.1 (67.4 to 72.5) and reduces diagonal attention by an additional 7.4%. The combined framework achieves the strongest performance and the most significant reduction in diagonal attention, confirming that the two methods target different aspects of the inertia problem and are not redundant.

---

> > ### Author Rebuttal · Reviewer_87nX · 2026-04-04
> >
> > The questions are fully solved.

---

### Decision · Program_Chairs · 2026-04-30

**Decision:**

Accept (regular)

**Comment:**

This paper identifies **conversational inertia** in multi-turn agents: a failure mode where models over-attend to their own previous responses. The paper and proposes two effective solutions: Context Preference Learning (CPL) and Clip Context. Its strengths lie in the novel, attention-based diagnosis of self-imitation bias and the practical, KV-cache-compatible nature of the methods, which demonstrate performance gains across several benchmarks.

During the rebuttal, the authors addressed reviewer concerns by providing comparisons against relevant baselines (e.g., StreamingLLM, SeCoM), clarifying the complementary nature of CPL and Clip, and providing hyperparameter heuristics. All reviewers have indicated that their concerns are fully resolved, and there is a clear consensus that the paper's problem formulation and empirical results justify an Accept recommendation.